# Credal Deep Ensembles for Uncertainty Quantification

**Kaizheng Wang**[1,4,*] **Fabio Cuzzolin**[3] **Shireen Kudukkil Manchingal**[3]
**Keivan Shariatmadar**[2,4] **David Moens**[2,4] **Hans Hallez**[1]
[1]KU Leuven, Department of Computer Science, DistriNet
[2]KU Leuven, Department of Mechanical Engineering, LMSD
[3]Oxford Brookes University, Visual Artificial Intelligence Laboratory
[4]Flanders Make@KU Leuven
{kaizheng.wang, keivan.shariatmadar, david.moens, hans.hallez}@kuleuven.be
{fabio.cuzzolin, 19185895}@brookes.ac.uk

## Abstract

This paper introduces an innovative approach to classification called *Credal Deep Ensembles* (CreDEs), namely, ensembles of novel *Credal-Set Neural Networks* (CreNets). CreNets are trained to predict a lower and an upper probability bound for each class, which, in turn, determine a convex set of probabilities (credal set) on the class set. The training employs a loss inspired by distributionally robust optimization which simulates the potential divergence of the test distribution from the training distribution, in such a way that the width of the predicted probability interval reflects the 'epistemic' uncertainty about the future data distribution. Ensembles can be constructed by training multiple CreNets, each associated with a different random seed, and averaging the outputted intervals. Extensive experiments are conducted on various out-of-distributions (OOD) detection benchmarks (CIFAR10/100 vs SVHN/Tiny-ImageNet, CIFAR10 vs CIFAR10-C, ImageNet vs ImageNet-O) and using different network architectures (ResNet50, VGG16, and ViT Base). Compared to Deep Ensemble baselines, CreDEs demonstrate higher test accuracy, lower expected calibration error, and significantly improved epistemic uncertainty estimation.

## 1 Introduction

The quantification of the uncertainty associated with neural network predictions has recently attracted increasing attention, to enhance the reliability and robustness of neural networks. Researchers agree to distinguish *aleatory uncertainty* (AU) from *epistemic uncertainty* (EU): the former arises from the inherent randomness, e.g., data noise, and is irreducible. The latter is caused by a lack of knowledge about the process which generates the data, due to the limited availability of training data, and is reducible [1, 36]. Effective EU quantification is beneficial for out-of-distribution (OOD) detection [30, 54] and can contribute to a variety of safety-critical applications, including autonomous driving [21], medical diagnosis [44], flood uncertainty estimation [10], structural health monitoring [72].

In classification, standard neural networks (SNNs) whose predictions amount to single probability distributions are unable to account for epistemic uncertainty, because a single distribution assumes precise knowledge about the dependency between inputs and outputs. To properly capture the EU, the network's outcome needs to express the uncertainty about a prediction's uncertainty itself [35, 62].

The most well-known approach to EU quantification in deep learning leverages Bayesian neural networks (BNNs) [7, 22, 38]. BNNs model network parameters as distributions and thus predict a 'second-order' distribution (i.e., a distribution of distributions) [36], although in practice predictions

---

[*]Corresponding author.

38th Conference on Neural Information Processing Systems (NeurIPS 2024).

are often generated by running the network on sample parameters extracted from a posterior. While efficient training techniques (such as sampling [33, 57] and variational inference [7, 22]) have been developed to mitigate their complexity, practical challenges persist for BNNs, including the scaling to large datasets, handling complex network architectures, and real-time applicability [54].

An alternative approach, called Deep Ensembles (DEs), handles uncertainty quantification by aggregating multiple individually-trained SNNs [43], so that predictions amount to finite sets of probability distributions over the classes. DEs, often viewed as an approximation of Bayesian model averaging (BMA) [38], have become a powerful baseline for uncertainty estimation [2, 25, 53, 54, 60]. However, empirical evidence [2] suggests that DEs could yield relatively low-quality estimates of epistemic uncertainty. Further, DEs lack a sound theoretical justification [27, 47].

Credal inference [13, 36, 62] (which predicts convex sets of probability distributions or *credal sets* [46] on the target space) can provide an alternative way of quantifying epistemic uncertainty. Credal representations have been extensively studied within the broader field of machine learning, including, e.g., the naive credal classifier [14], the notion of credal network [13] or credal random forest classification [65]. 'Imprecise' BNNs have been recently introduced which model both network weights and predictions as credal sets [8]. While these models demonstrate robustness in Bayesian sensitivity analysis, their computational complexity is on a par with that of ensembles of BNNs, significantly limiting their practical applicability.

**Novelty and Main Contributions**. This paper presents an innovative approach to classification tasks called *Credal Deep Ensembles* (CreDEs), ensembles of novel *Credal-Set Neural Networks* (CreNets), aiming to improve EU quantification in the framework of credal inference. At the cost of merely doubling the number of output nodes compared to classical SNNs, CreNets are trained to predict a lower and an upper probability bound for each class rather than a single probability value. Such probability intervals over classes thus efficiently determine a prediction in the form of a credal set. The training strategy is inspired by Distributionally Robust Optimization [42, 55, 61], which simulates the potential divergence of the test distribution from the training distribution. As a result, the width of the predicted probability interval reflects the 'epistemic' uncertainty about the future data distribution. Adopting an ensemble strategy, CreDEs derive the final prediction by averaging the probability intervals outputted by the members of the ensemble. A conceptual comparison between CreDEs and DEs is illustrated in Figure 1.

Extensive experimental validation is conducted on several OOD detection benchmarks, including CIFAR10/100 (ID) vs SVHN/Tiny-ImageNet (OOD), CIFAR10 (ID) vs CIFAR10-C (OOD), ImageNet (ID) vs ImageNet-O (OOD), and across different network architectures: ResNet50, VGG16 and Visual Transformer Base (ViT Base). Compared to traditional Deep Ensembles, our CreDEs achieve *higher test accuracy* and *lower expected calibration error* (ECE) on ID samples, and *significantly improve the quality of EU estimation*.

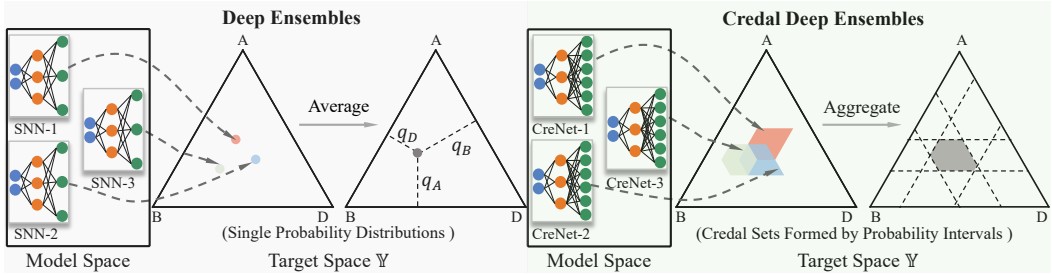

Figure 1: Comparison between the proposed Credal Deep Ensembles and traditional Deep Ensembles. The former aggregate a collection of credal set predictions from CreNets as the final (credal) prediction, whereas the latter average a set of single probability distributions from standard SNNs as the outcome. E.g., in the probability simplex [16] associated with the target space $\mathbb{Y} = \{A, B, D\}$ (the triangle in the figure), a probability vector $(q_A, q_B, q_D)$ is represented as a single point. For each CreNet, the predicted lower and upper probabilities of each class act as constraints (parallel lines) which determine a credal prediction (in gray). Single credal predictions are aggregated as in Sec. 2.4.

**Related Work** Besides BNNs, DEs, and credal inference, other 'second-order' uncertainty estimation approaches exist, such as Dirichlet-based methods [9, 48, 49, 50, 63], in which predictions are represented as Dirichlet distributions. One significant challenge for the latter is the absence of ground

truth labels. Although various loss functions have been proposed, these models' performance often diverges from theoretical EU assumptions [71]. Another rationale for the exclusion of Dirichlet-based approaches as baselines for our CreDE work is that such models often necessitate the inclusion of OOD data during training [48, 49, 56]. This challenges their practical adaptability, as it cannot guarantee their robustness against other forms of 'unseen' OOD data [71]. Moreover, a recent study [39] has shown that these methods often fail to capture the EU properly, making the resulting measures difficult to interpret quantitatively.

**Paper Outline** The remainder of this paper is structured as follows. Sec. 2 presents our CreNets and CreDEs in full detail. Sec. 3 describes the experimental validations and results. Sec. 4 summarizes our conclusions and future work. Appendices report mathematical proofs in §A, additional experiments in §B, implementation details in §C, the analysis of alternative ensemble strategies for CreDEs in §D, and further discussion on future work in §E (including achieving statistical guarantees using conformal learning and the framework's extension to regression), respectively.

## 2 Approach

The proposed Credal-Set Neural Network architecture and forward propagation are introduced in Sec. 2.1. CreNets' training procedure is discussed in Sec. 2.2. The class prediction and uncertainty quantification are discussed in Sec. 2.3. Credal Deep Ensembles are presented in Sec. 2.4.

### 2.1 Credal-Set Neural Networks

Architecturally, our CreNet design focuses only on the final classification layers, and can therefore be applied on top of any representation layers of neural network models. The final layers of a CreNet (Figure 2) first output a deterministic interval for each class, using for each class an output node associated with the interval midpoint $m$ and one associated with its half-length $h$, respectively (a total of $2C$ nodes for $C$ classes).

Let $\boldsymbol{z}$ be the input vector to the final layer. CreNets compute $\boldsymbol{m}$ and $\boldsymbol{h}$ (the vectors collecting interval midpoints and half-lengths for all classes) as:

$$\begin{aligned} \boldsymbol{m} &= g(\mathbf{W}_{1:C} \cdot \boldsymbol{z} + \boldsymbol{b}_{1:C}) \\ \boldsymbol{h} &= \zeta(\mathbf{W}_{C+1:2C} \cdot \boldsymbol{z} + \boldsymbol{b}_{C+1:2C}) \end{aligned} \quad (1)$$

where $\mathbf{W}_{1:C}, \boldsymbol{b}_{1:C}, \mathbf{W}_{C+1:2C}, \boldsymbol{b}_{C+1:2C}$ are the weights and biases associated with the first $C$ and the remaining $C$ nodes, respectively. Here $g(\cdot)$ is an arbitrary activation function and $\zeta(\cdot)$ denotes the Softplus function [79] that ensures the non-negativity of $\boldsymbol{h}$.

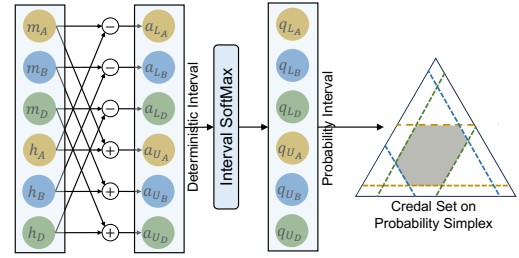

Figure 2: CreNet final layer structure for three classes.

The deterministic intervals associated with all classes, denoted as $[\boldsymbol{a}_L, \boldsymbol{a}_U] := \{[a_{L_i}, a_{U_i}]\}_{i=1}^C$, can then be obtained as $[\boldsymbol{a}_L, \boldsymbol{a}_U] = [\boldsymbol{m} - \boldsymbol{h}, \boldsymbol{m} + \boldsymbol{h}]$.

A proper mapping from such deterministic intervals $[\boldsymbol{a}_L, \boldsymbol{a}_U]$ to a collection of probability intervals $[\boldsymbol{q}_L, \boldsymbol{q}_U] := \{[q_{L_i}, q_{U_i}]\}_{i=1}^C$ for each class needs to ensure that $[\boldsymbol{q}_L, \boldsymbol{q}_U]$ satisfies the conditions:

$$q_{L_i} \leq q_{U_i} \ \forall i = 1, ..., C \quad \text{and} \quad \sum_{i=1}^C q_{L_i} \leq 1 \leq \sum_{i=1}^C q_{U_i}. \quad (2)$$

The former condition guarantees a proper $[q_{L_i}, q_{U_i}]$ for each class. The latter enables the resulting collection of probability intervals to determine a non-empty credal set, $\mathbb{Q}$, as follows [52]:

$$\mathbb{Q} = \{\boldsymbol{q} \mid q_i \in [q_{L_i}, q_{U_i}]; \sum_{i=1}^C q_i = 1\}. \quad (3)$$

The probability vectors in $\mathbb{Q}$ meet the normalization condition, and their probability value per class is constrained by the probability intervals (Eq. (2)).

Traditional SoftMax activation cannot ensure that the convexity conditions in Eq. (2) are met when computing $[\boldsymbol{q}_L, \boldsymbol{q}_U]$ using $\boldsymbol{q}_L = \text{SoftMax}(\boldsymbol{a}_L)$ and $\boldsymbol{q}_U = \text{SoftMax}(\boldsymbol{a}_U)$, respectively. A toy example

is given in Appendix §A. Therefore, we employ Interval SoftMax activation as proposed in [75] to compute $[\boldsymbol{q}_L, \boldsymbol{q}_U]$ from $[\boldsymbol{a}_L, \boldsymbol{a}_U]$, as follows:

$$q_{Li} = \frac{\exp(a_{L_i})}{\exp(a_{L_i}) + \sum_{k \neq i} \exp(\frac{a_{U_k} + a_{L_k}}{2})}, \quad q_{Ui} = \frac{\exp(a_{U_i})}{\exp(a_{U_i}) + \sum_{k \neq i} \exp(\frac{a_{U_k} + a_{L_k}}{2})}, \tag{4}$$

where $q_{L_i}$ and $q_{U_i}$ are the lower and upper probability bound for the $i^{th}$ class, respectively. As proven in Appendix §A, the probability intervals generated by the Interval SoftMax duly satisfy Eq. (2).

## 2.2 Training Procedure

The rationale for the training of a CreNet is for the predicted lower and upper bounds (Eq. (4)) to the probability of the classes, $\boldsymbol{q}_L$ and $\boldsymbol{q}_U$, to express the epistemic uncertainty (induced by the limited size and variability of the training set) about how different the distribution of future test data may be from that of the training data.

To this extent, we designed a composite loss function with two components: one, which applies classical cross entropy to the upper probability vector, encourages the latter to optimistically assume that test data distribution will be similar. The other, inspired by Distributionally Robust Optimization (DRO) [42, 55, 61], pushes the lower probability to reflect a 'pessimistic' stance on future distributional divergence. The width of the resulting interval will thus reflect the epistemic uncertainty associated with the prediction.

We first contrast the classical training strategy with that of Distributionally Robust Optimization in Sec. 2.2.1. We then delve into the design and implementation of our CreNet loss in Sec. 2.2.2.

### 2.2.1 Classical and DRO Training Strategy

**Vanilla Strategy** Given a training set $\mathbb{D} = \{\boldsymbol{x}_n, \boldsymbol{t}_n\}_{n=1}^N$, the conventional neural network training process aims to solve the following optimization problem

$$\underset{\boldsymbol{\theta} \in \Theta}{\text{minimize}} \left\{ \frac{1}{N} \sum\nolimits_{n=1}^N \mathcal{L}((\boldsymbol{x}_n, \boldsymbol{t}_n), \boldsymbol{\theta}) \right\}, \tag{5}$$

where $\boldsymbol{\theta}$ denotes the model's trainable parameters in the space $\Theta$ and $\mathcal{L}$ denotes an arbitrary loss function. The underlying assumption is that the training and test distributions are identical. As a result, the trained network serves as an empirical risk minimizer [36]. However, this ideal assumption often results in over-optimistic predictions because the test observations may, in practice, significantly differ from the training data [34].

**DRO Strategy** In contrast to the vanilla strategy, the objective of DRO [6, 20] is to minimize the worst-case expected risk ($R(\boldsymbol{\theta})$) over an uncertain set of distributions $\mathcal{U}$, as follows:

$$\underset{\boldsymbol{\theta} \in \Theta}{\text{minimize}} \left\{ R(\boldsymbol{\theta}) \doteq \sup_{U \in \mathcal{U}} \mathbb{E}_{(\boldsymbol{x}, \boldsymbol{t}) \sim U} \mathcal{L}((\boldsymbol{x}, \boldsymbol{t}), \boldsymbol{\theta}) \right\}, \tag{6}$$

in which $\mathbb{E}$ is the expectation operation. In practice, a *group DRO* setting [59] is adopted in which the training distribution $P$ is assumed to be a mixture of $m$ groups $P_g$, indexed by $g \in \mathcal{G} = \{1, 2, ..., m\}$. Because the optimum of a linear program is attained at a vertex, the worst-case risk in Eq. (6) is equivalent to a maximum over the expected loss of each group, as follows:

$$R(\theta) = \underset{g \in \mathcal{G}}{\text{maximize}} \, \mathbb{E}_{(\boldsymbol{x}, \boldsymbol{t}) \sim P_g} \mathcal{L}((\boldsymbol{x}, \boldsymbol{t}), \boldsymbol{\theta}). \tag{7}$$

In practice, the group DRO model minimizes the empirical worst-group risk $\hat{R}(\theta)$, namely:

$$\underset{\boldsymbol{\theta} \in \Theta}{\text{minimize}} \left\{ \hat{R}(\boldsymbol{\theta}) \doteq \underset{g \in \mathcal{G}}{\text{maximize}} \, \mathbb{E}_{(\boldsymbol{x}, \boldsymbol{t}) \sim \hat{P}_g} \mathcal{L}((\boldsymbol{x}, \boldsymbol{t}), \boldsymbol{\theta}) \right\}, \tag{8}$$

where $\hat{P}_g$ is the empirical distribution of the $g$-th group of training points. Therefore, group DRO learns models with good worst-group training loss across groups [59]. One special form of *group DRO* is *adversarially reweighted learning* [42], which consists of a minimax game between a learner and adversary. The learner optimizes for the main classification task and aims to learn the best

parameters $\boldsymbol{\theta}$ that minimize the expected loss. In contrast, the adversary maximizes the expected loss by making an adversarial assignment of weights $w_n$, collected in a vector $\boldsymbol{w}$. Consequently, the training optimization problem assumes the form

$$\underset{\boldsymbol{\theta} \in \Theta}{\text{minimize}} \left\{ \underset{\boldsymbol{w} \in \mathbb{S}}{\text{maximize}} \frac{1}{N} \sum_{n=1}^{N} w_n \cdot \mathcal{L}((\boldsymbol{x}_n, \boldsymbol{t}_n), \boldsymbol{\theta}) \right\}, \tag{9}$$

where the set $\mathbb{S}$ of weight vectors varies across different implementations [42, 55, 61].

### 2.2.2 CreNet Loss Design and Implementation

**Design** As anticipated, the CreNet training process applies the vanilla training strategy to the upper probability prediction vector $\boldsymbol{q}_U$ (Eq. (5)), and the DRO strategy to the lower probability prediction $\boldsymbol{q}_L$ (Eq. (9)). The resulting overall loss function has a composite structure

$$\mathcal{L}_{\text{CreNet}} := \underbrace{\frac{1}{N} \sum_{n=1}^{N} \text{CE}(\boldsymbol{q}_{U_n}, \boldsymbol{t}_n)}_{\text{Vanilla Component}} + \underbrace{\underset{\boldsymbol{w} \in \mathbb{S}}{\text{maximize}} \frac{1}{N} \sum_{n=1}^{N} w_n \cdot \text{CE}(\boldsymbol{q}_{L_n}, \boldsymbol{t}_n)}_{\text{DRO Component}}, \tag{10}$$

where CE denotes the classical cross-entropy loss function used in classification. Given a predicted discrete probability vector $\boldsymbol{q}$ and the ground-truth label $\boldsymbol{t}$, CE is defined as: $\text{CE} := -\sum_{k}^{C} t_k \cdot \log_2 q_k$. The vanilla component is applied to the upper probability vector $\boldsymbol{q}_U$ because such a loss takes the training distribution at face value and is thus more likely to encourage 'optimistic' (overconfident) or 'upper bound' predictions for the class scores. The DRO component is computed on the lower probability vectors $\boldsymbol{q}_L$, as it weighs training outliers to simulate future differences in data distribution at test time, encouraging 'pessimistic' or 'lower bound' class score predictions. Thus, the width of the resulting probability interval will reflect the uncertainty associated with the model's ignorance of how much the future test distribution will differ from the train distribution, using the boundary/outlier cases observed at training time to guess what the uncertainty on future test cases will be.

**Cross-Entropy of Lower/Upper Probability Vectors** Please note that in Eq. (10), the CE is applied to lower/upper probability vectors, which are not (normalized) probability vectors. However, as the ground truth (label) vector $\boldsymbol{t}$ equals 1 for the true class $j^*$ and 0 for all the other elements, calculating $\text{CE}(\boldsymbol{q}, \boldsymbol{t})$ for any predicted probability vector $\boldsymbol{q}$ reduces to $-\log_2 q(j^*)$. Consequently, all probability vectors with the same component for the true class will generate the same CE for that sample.

The consequence for CreNet training is that feeding a lower (upper) probability vector $\boldsymbol{q}_L$ ($\boldsymbol{q}_U$) to Eq. (10) is equivalent to computing the CE with *any one* of the probability vectors in the credal prediction (Eq. (3)) whose probability for the true class $j^*$ equals the lower (upper) probability value there. It can be shown that these form one of the 'faces' of the boundary of the credal set.

Importantly, because of the functional structure of Interval SoftMax activation (Eq. (4)), upper and lower probability vectors are not computed independently, but are correlated. Thus, they are minimized together via the total loss (Eq. (10)), with the DRO component also influencing the upper probability $\boldsymbol{q}_U$, driving the solution away from the trivial one (all-ones upper probability vectors).

**Implementation** As the $\mathbb{S}$ of weight vectors in Eq. (10) varies across different implementations [42, 55, 61] and estimating $\boldsymbol{w}$ in Eq. (10) is not straightforward when using batch-wise training [34], we resort to a simpler heuristic proposed by [34]. For each training batch, only the $\delta \in [0.5, 1)$ portion of samples with the highest cross-entropy with the lower probability vector ($\text{CE}(\boldsymbol{q}_{L_n}, \boldsymbol{t}_n)$) are selected to compute the DRO component of the loss. As a result, $w_n > 1$ is implicitly set in Eq. (10) for selected samples while $w_n = 0$ for deselected samples.

The underlying rationale is the following. Within a batch of samples, those instances that demonstrate high losses are identified as 'hard-to-learn' samples, essentially representing the 'minority group' within a training dataset [34]. Setting a value $\delta$ thus identifies what fraction of the training points is chosen to represent potential future domain shifts at test time. A smaller $\delta$ signifies a more cautious approach, in which even a few training outliers can indicate future challenges.

The lower bound to the design range for $\delta$ is 0.5 because we empirically observed that values of $\delta < 0.5$ may destabilize the training process, as a too-large averaged loss is returned for backpropagation. When $\delta$ approaches 1, the data distribution of the samples considered by the vanilla and the DRO components of the loss becomes similar, implicitly assuming a less pronounced divergence between train and test distributions. The corresponding predicted probability intervals become narrower. If $\delta$

were theoretically set to 1, all samples would be selected for backpropagation, implying that $w_n = 1$ for any $n$ in Eq. (10). Consequently, the loss in Eq. (10) would be the sum of the vanilla component on $q_U$ and the vanilla component on $q_L$. Empirically, we observed that this leads to a collapse of the upper and lower probability bounds to single values.

The implementation of the CreNet training procedure is shown in Algorithm 1.

---
**Algorithm 1** CreNet Training Procedure

---
**Input:** Training dataset $\mathbb{D} = \{\boldsymbol{x}_n, \boldsymbol{t}_n\}_{n=1}^N$; Portion of samples per batch $\delta \in [0.5, 1)$; Batch size $\eta$
**while** *enable training* **do**
    1. Compute $\text{CE}(\boldsymbol{q}_{U_n}, \boldsymbol{t}_n)$ and $\text{CE}(\boldsymbol{q}_{L_n}, \boldsymbol{t}_n)$ for each sample
    2. Sort the sample indices $(m_1, ..., m_\eta)$ in descending order of $\text{CE}(\boldsymbol{q}_{L_n}, \boldsymbol{t}_n)$
    3. Define $\eta_\delta = \lfloor \delta\eta \rfloor$
    4. Minimize $\mathcal{L}_{\text{CreNet}} = \frac{1}{\eta} \sum_{n=1}^{\eta} \text{CE}(\boldsymbol{q}_{U_n}, \boldsymbol{t}_n) + \frac{1}{\eta_\delta} \sum_{j=1}^{\eta_\delta} \text{CE}(\boldsymbol{q}_{L_{m_j}}, \boldsymbol{t}_{m_j})$
**end while**

---

### 2.3 Class Prediction and Uncertainty Quantification

**Class Prediction** For the class prediction we employ the 'maximin' and 'maximax' criteria [66]:

$$\hat{i}_{\min} := \underset{i}{\arg\max}\, q_{L_i}^*; \quad \hat{i}_{\max} := \underset{i}{\arg\max}\, q_{U_i}^*, \tag{11}$$

which output (respectively) the class indices with the highest lower and upper reachable probability ($q_{L_i}^*$ and $q_{U_i}^*$) within the same credal set induced by the predicted lower and upper probabilities $q_{L_i}, q_{U_i}$. Figure 3 illustrates how the lower and upper probabilities $q_{L_i}, q_{U_i}$ that determine the credal set $\mathbb{Q}$ may differ from the probabilities $q_{L_i}^*$ and $q_{U_i}^*$ *actually reachable* for each class within $\mathbb{Q}$. The reachable lower and upper probabilities for class $i$ can be easily obtained as follows [17]:

$$q_{U_i}^* = \min\left(q_{U_i}, 1 - \sum_{j \neq i} q_{L_j}\right), q_{L_i}^* = \max\left(q_{L_i}, 1 - \sum_{j \neq i} q_{U_j}\right). \tag{12}$$

**Uncertainty Quantification** Given a credal set prediction, upper and lower entropies generalizing Shannon's entropy, denoted as $\overline{H}(\mathbb{Q})$ and $\underline{H}(\mathbb{Q})$, can be defined which may serve as measures for TU and AU, respectively [3, 36].

Computing $\overline{H}(\mathbb{Q})$ boils down to solving the following optimization problem:

$$\overline{H}(\mathbb{Q}) = \text{maximize} \sum_{i=1}^{C} -q_i \cdot \log_2 q_i \quad \text{s.t.} \quad q_{L_i}^* \leq q_i \leq q_{U_i}^* \,\forall i \quad \text{and} \quad \sum_{i=1}^{C} q_i = 1. \tag{13}$$

This seeks the highest entropy value of a probability distribution within the predicted credal set $\mathbb{Q}$. $\underline{H}(\mathbb{Q})$, for which $\text{maximize}$ is replaced by $\text{minimize}$, searches for the minimal such entropy. Such optimization problems can be addressed using a standard solver, e.g., the SciPy optimization package [73]. Epistemic uncertainty can then be quantified as $\overline{H}(\mathbb{Q}) - \underline{H}(\mathbb{Q})$ [36].

**Computational Complexity Reduction** To reduce the computational complexity of Eq. (13) for a large value of $C$ (e.g., $C = 1000$), we propose an original approach called *Probability Interval Dimension Reduction* (PIDR) in Algorithm 2. This method first identifies the $K-1$ classes with the highest lower probability values, then merges the remaining elements into a single class with the associated upper and lower probability calculated using Eq. (12). Consequently, the dimension of the probability interval is reduced from $C$ to $K$.

### 2.4 Credal Deep Ensembles

Inspired by conventional DEs [43], the final step of our approach is to introduce *Credal Deep Ensembles* (CreDEs). CreDEs aggregate $M$ individually trained CreNets and predict the aggregated probability intervals, denoted as $[\tilde{\boldsymbol{q}}_L^*, \tilde{\boldsymbol{q}}_U^*]$, as follows:

$$\tilde{\boldsymbol{q}}_L^* = \frac{1}{M} \sum_{m=1}^{M} \boldsymbol{q}_{L_m}^*, \quad \tilde{\boldsymbol{q}}_U^* = \frac{1}{M} \sum_{m=1}^{M} \boldsymbol{q}_{U_m}^*, \tag{14}$$

where $[\boldsymbol{q}_{L_m}^*, \boldsymbol{q}_{U_m}^*]$ is the set of reachable probability intervals predicted by the $m$-th CreNet. Eq. (20) in Appendix D proves that $[\tilde{\boldsymbol{q}}_L^*, \tilde{\boldsymbol{q}}_U^*]$ satisfies the convexity condition in Eq. (2) for constructing a

**Algorithm 2** Probability Interval Dimension Reduction Algorithm

**Input:** $[\boldsymbol{q}_L^*, \boldsymbol{q}_U^*]$; Chosen number of classes $K$
**Output:** Reduced-dimensional probability intervals $[\boldsymbol{r}_L, \boldsymbol{r}_U]$
**1.** Index vector of $\boldsymbol{q}_L^*$ in descending order: $\boldsymbol{l} \leftarrow \operatorname{argsort}(\boldsymbol{q}_L^*)$
**2.** Define the upper and lower probability per selected class:
$r_{L_j} \leftarrow q_{L_{l_j}}^*, r_{U_j} \leftarrow q_{U_{l_j}}^*$ for $j = 1, ..., K-1$
**3.** Define upper and lower probability for deselected classes:
$r_{L_K} \leftarrow \max(1 - \sum_{i=l_C}^{l_C} q_{U_i}^*, \sum_{j=1}^{K-1} r_{L_j})$;
$r_{U_K} \leftarrow \min(1 - \sum_{i=l_K}^{l_C} q_{L_i}^*, \sum_{j=1}^{K-1} r_{U_j})$

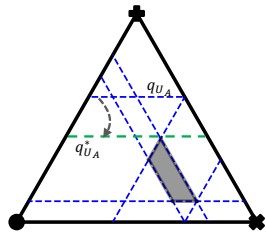

Figure 3: If intervals are redundant, some of the (e.g.) upper probabilities $q_{U_A}$ may not be actually reachable in the credal set that results from the intersection of all interval constraints.

non-empty credal set. Therefore, class prediction and uncertainty estimation as described in Sec. 2.3 apply to CreDEs. We discuss the rationale for averaging strategy and the alternative ensemble approaches for CreDEs in Appendix §D.

## 3  Experimental Validation

**Setup** We assessed CreDEs through OOD detection benchmarks across various dataset pairings (ID vs OOD samples), including CIFAR10 [41]/CIFAR100 [40] vs SVHN [31]/Tiny-ImageNet [45], CIFAR10 vs CIFAR10-C [29], ImageNet [18] vs ImageNet-O [31]. We trained 15 CreNets (using $\delta = 0.5$) and SNNs on the ResNet50 architecture [28] starting from different random seeds, using the training set as per the ID dataset in the pair. Following this, we constructed 15 different CreDEs and DEs, respectively, by randomly selecting five members from the pool of 15 trained models. The same ensemble member lists are used for both DEs and CreDEs, with each ensemble strictly guaranteed to be distinct. More details are given in Appendix §C. Codes are available at `https://gitlab.kuleuven.be/m-group-campus-brugge/distrinet_public/credal-deep-ensembles.git`.

**Uncertainty Quantification in DEs** Total uncertainty (TU) can be quantified in DEs via the Shannon entropy ($H$) of the averaged predicted distribution. The AU, on the other hand, can be obtained by averaging the entropies of the predictions of each ensemble member [2, 36]. Namely,

$$\text{TU} := H(\tilde{\boldsymbol{q}}) = H(\tfrac{1}{M} \sum_{m=1}^{M} \boldsymbol{q}_m), \quad \text{AU} := \tilde{H}(\boldsymbol{q}) = \tfrac{1}{M} \sum_{m=1}^{M} H(\boldsymbol{q}_m), \tag{15}$$

where $M$ is the number of networks, $\tilde{\boldsymbol{q}}$ and $\boldsymbol{q}_m$ denote the average probability vector and the single probability vector of the $m$-th SNN model, respectively. The level of epistemic uncertainty, representing an approximation of mutual information [36], can be obtained as $\text{EU} := H(\tilde{\boldsymbol{q}}) - \tilde{H}(\boldsymbol{q})$.

**Test Accuracy and ECE on ID Samples** We evaluated the test accuracy and expected calibration error (ECE) [24, 58] of CreDEs-5 and DEs-5 on the test set of each ID dataset. A lower ECE value signifies a closer alignment between the model's confidence scores and the true probabilities of the events. Since ECE is designed for a singular probability vector, we implemented a compromise calculation as follows. Suppose our model predicts the class indices $k$ and $j$ when using the $\hat{i}_{\min}$ and $\hat{i}_{\max}$ criteria, respectively, ECE values are then computed based on the associated lower $q_{L_k}^*$ and upper $q_{U_j}^*$ reachable probabilities in the credal set.

Table 1 reports the test accuracy and ECE for DEs-5 and CreDEs-5 on the various datasets, indicating that our CreDEs-5 achieved *higher test accuracy* and *lower ECE* on ID samples. Note that employing the $\hat{i}_{\min}$ prediction showed higher ECE on the challenging ImageNet dataset. This is likely because the strategy, selecting the class with the highest lower reachable probability, is a conservative one.

Table 1: Test accuracy (%, ↑) and ECE (↓) of DEs-5 and CreDEs-5 using CIFAR10, CIFAR100, and ImageNet as ID datasets over 15 runs. The better performance is marked in bold.

| | | CIFAR10 | | CIFAR100 | | ImageNet | |
|---|---|---|---|---|---|---|---|
| | | Test Accuracy | ECE | Test Accuracy | ECE | Test Accuracy | ECE |
| DEs-5 | | 93.32±0.13 | 0.0131±0.0010 | 75.80±0.28 | 0.0392±0.0027 | 77.92±0.02 | 0.2415±0.0009 |
| CreDEs-5 (Ours) | $\hat{i}_{\min}$ | **93.75±0.11** | **0.0092±0.0016** | **79.54±0.21** | **0.0366±0.0025** | **78.41±0.02** | 0.5930±0.0006 |
| | $\hat{i}_{\max}$ | **93.74±0.11** | **0.0108±0.0017** | **79.65±0.19** | **0.0268±0.0023** | **78.51±0.02** | **0.1685±0.0004** |

Table 2: OOD detection AUROC and AUPRC performance (%, ↑) between CreDEs-5 and DEs-5 based on ResNet50 using EU as uncertainty metrics on CIFAR10/100 vs. SVHN/Tiny-ImageNet and ImageNet vs. ImageNet-O. Results are averaged over 15 runs. Best results in bold.

| ID Samples | | CIFAR10 | | | CIFAR100 | | | ImageNet | |
|---|---|---|---|---|---|---|---|---|---|
| OOD Samples | | SVHN | | Tiny-ImageNet | | SVHN | | Tiny-ImageNet | ImageNet-O |
| Performance Indicator | AUROC | AUPRC | AUROC | AUPRC | AUROC | AUPRC | AUROC | AUPRC | AUROC | AUPRC |
| DEs-5 $H(\bar{q})-\tilde{H}(q)$ | 89.58±0.93 | 92.29±1.00 | 86.87±0.20 | 83.02±0.16 | 73.83±1.97 | 84.96±1.25 | 78.80±0.20 | 74.68±0.27 | 65.03±0.53 | 62.77±0.38 |
| CreDEs-5 $\overline{H}(\mathbb{Q})-\underline{H}(\mathbb{Q})$ | **96.55±0.25** | **98.17±0.17** | **88.10±0.26** | **87.85±0.35** | **78.55±1.15** | **86.57±0.65** | **82.54±0.26** | **77.60±0.44** | **67.82±0.06** | **62.80±0.12** |

**EU Quantification for OOD Detection** It is our hypothesis that OOD data express a higher EU. Hence, we can use a better EU quantification as the means to improve the OOD detection [54]. Thus, superior OOD detection performance provides compelling evidence of enhanced uncertainty estimation quality. For the OOD detection performance assessment, we employed AUROC (Area Under the Receiver Operating Characteristic curve) and AUPRC (Area Under the Precision-Recall curve) scores. AUROC captures true and false positive rates, while AUPRC assesses precision and recall trade-offs, offering valuable insights into model effectiveness across various confidence levels. When calculating $\overline{H}(\mathbb{Q})$ and $\underline{H}(\mathbb{Q})$ in the ImageNet vs ImageNet-O experiment, we employed our PIDR Algorithm 2 with $K = 20$. Table 2 reports the OOD detection performance of

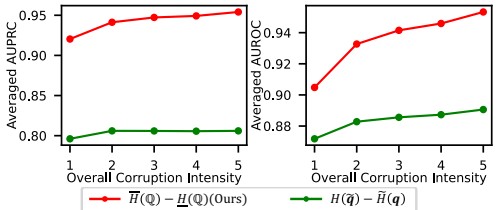

Figure 4: OOD detection (CIFAR10 vs CIFAR10-C) over increased corruption intensity.

CreDEs-5 and DEs-5 in the CIFAR10/CIFAR100 vs SVHN/Tiny-ImageNet, and ImageNet vs ImageNet-O settings. As the CIFAR10-C dataset contemplates data from CIFAR10 corrupted in 15 distinct ways, each with 5 different intensities, Figure 4 presents averaged AUROC and AUPRC scores for OOD detection on CIFAR10 vs CIFAR10-C across types of corruption, against the intensity of corruption. Table 2 and Figure 4 confirm CreDEs-5's superior OOD detection performance over DEs-5. This indicates the effectiveness of CreDEs in improving the EU quantification quality, using $\overline{H}(\mathbb{Q})-\underline{H}(\mathbb{Q})$ as the uncertainty measures.

**Qualitative Evaluation** Due to the high dimensionality, visualizing or directly computing the size of the credal set becomes challenging as $C$ increases. Consequently, we indirectly evaluate whether our CreDEs consistently generate nearly Dirac credal sets as predictions through the maximum attainable upper bound probability of the prediction. The closer this probability is to 1, the more it approximates a Dirac credal set. Figure 5 shows the results of ResNet50-based CreDEs-5 for the CIFAR10, SVHN, and Tiny-ImageNet datasets. It verifies that our method does not consistently generate nearly Dirac credal sets, especially for OOD samples. For CIFAR10, a substantial proportion of (but not all) the credal sets are quasi-Dirac. This observation is reasonable as it is consistent with the high test accuracy of CreDEs and the low ECE reported in Table 1.

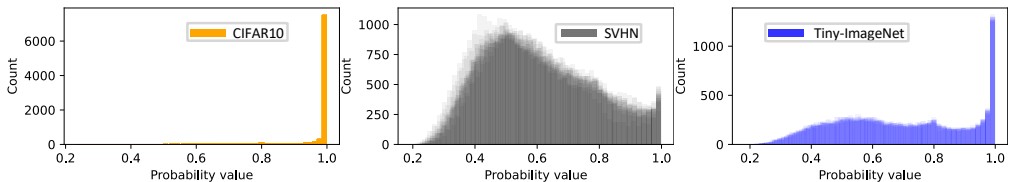

Figure 5: Maximum reachable upper probability $\max(q^*_{U_1}, ..., q^*_{U_C})$ per sample from 15 runs.

Additionally, Figure 6 shows the reliability diagram[24] of the ResNet50-based DEs-5 and CreDEs-5 on the CIFAR10 dataset, demonstrating better calibration performance of our CreDEs. Figure 7 showcases the EU estimation plots for these models. Although the EU estimates for DEs-5 and CreDEs-5 are not directly comparable due to differing representations, CreDEs-5 demonstrates significantly higher EU estimates for OOD samples, as observed qualitatively.

**Ablation Study on Various Network Architectures** We also performed an ablation study on network backbones different from ResNet50, including VGG16 [67] and Vision Transformer Base (ViT Base)

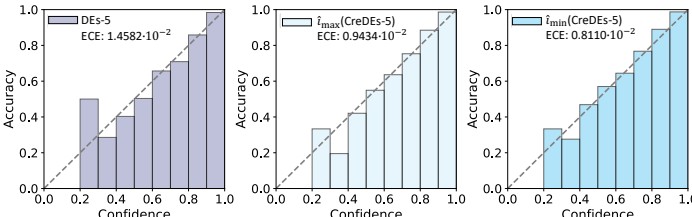
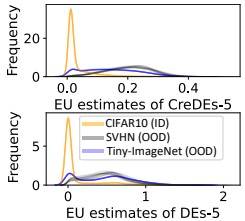

Figure 6: Reliability diagram of ResNet50-based DEs-5 and CreDEs-5 (using $\hat{i}_{min}$ and $\hat{i}_{max}$, respectively) on CIFAR10.

Figure 7: EU estimates comparison of ResNet50-based models.

[77]. Table 3 reports the test accuracy and ECE of CreDEs-5 and DEs-5 on the CIFAR10 test set (representing ID samples) and the OOD detection performance on CIFAR10 vs SVHN/Tiny-ImageNet. Figure 8 compares OOD detection performance in the CIFAR10 vs CIFAR10-C setting against the intensity of corruption, using both AUPRC and AUROC as metrics. The results consistently demonstrate that CreDEs achieve *higher test accuracy*, *lower ECE*, and *significantly improved epistemic uncertainty estimation*, leading to enhanced OOD detection performance.

Table 3: Test accuracy (%, ↑) and ECE (↓) of DEs-5 and CreDEs-5 on CIFAR10 as ID dataset (left). AUROC and AUPRC scores (%, ↑) for OOD detection on CIFAR10 vs SVHN/Tiny-ImageNet (right). Results averaged over 15 runs. The Best results are in bold.

| | | | CIFAR10 (ID) | | | CIFAR10 vs SVHN | | CIFAR10 vs Tiny-ImageNet | |
| --- | --- | --- | --- | --- | --- | --- | --- | --- | --- |
| | | | Test Accuracy | ECE | | AUROC | AUPRC | AUROC | AUPRC |
| VGG16 | DEs-5 | | 85.53±0.10 | 0.0815±0.0011 | $H(\bar{q})-\tilde{H}(q)$ | 82.19±0.82 | 87.52±0.81 | 78.58±0.15 | 73.28±0.23 |
| | CreDEs-5 (Ours) | $\hat{i}_{min}$ | 87.94±0.11 | 0.0203±0.0014 | $\overline{H}(\mathbb{Q})-\underline{H}(\mathbb{Q})$ | 87.68±0.73 | 93.47±0.57 | 82.56±0.28 | 80.81±0.52 |
| | | $\hat{i}_{max}$ | 87.92±0.11 | 0.0611±0.0012 | | | | | |
| ViT Base | DEs-5 | | 90.43±0.97 | 0.0181±0.0019 | $H(\bar{q})-\tilde{H}(q)$ | 77.71±1.67 | 88.73±0.32 | 82.27±0.79 | 78.85±0.81 |
| | CreDEs-5 (Ours) | $\hat{i}_{min}$ | 93.60±0.40 | 0.0107±0.0014 | $\overline{H}(\mathbb{Q})-\underline{H}(\mathbb{Q})$ | 88.57±2.08 | 93.24±1.25 | 88.73±0.32 | 87.84±0.52 |
| | | $\hat{i}_{max}$ | 93.59±0.39 | 0.0104±0.0012 | | | | | |

Figure 8: OOD detection on CIFAR10 vs CIFAR10-C against increased corruption intensity, using VGG16 and ViT Base as backbones.

**Ablation Study on Hyperparameter $\delta$ for CreNet Training** In our main evaluation, we set by default $\delta = 0.5$ to reflect a balanced assessment of the train-test divergence and show how such a value allows our model to outperform the baselines. Table 4 reports the test accuracy and OOD detection performance (using EU estimates) of CreDEs-5 under various values of $\delta$. The ablation findings verify the robustness of Cre-DEs across hyperparameter setups and indicate the $\delta = 0.5$ might be too pessimistic a choice in CIFAR10 settings. Performance peaks at $\delta = 0.875$ in most cases, implying that $\delta = 0.875$ may provide the 'optimal' estimate of how test and train sets diverge for CIFAR10. One possible way to find the 'best' $\delta$ in practice is to conduct standard cross-validation on specific test scenarios. However, the method is not particularly sensitive to this hyperparameter. Per-

Table 4: Test accuracy (%, ↑) and OOD detection performance (%, ↑) of CreDEs-5 using various $\delta$. Results are averaged over 15 runs.

| $\delta$ | | 0.5 | 0.625 | 0.75 | 0.875 | 0.9375 | 0.96875 |
| --- | --- | --- | --- | --- | --- | --- | --- |
| Test Accuracy (CIFAR10) | $\hat{i}_{max}$ | 93.74 | 94.54 | 94.47 | 94.57 | 93.88 | 93.99 |
| | $\hat{i}_{min}$ | 93.75 | 94.55 | 94.47 | 94.56 | 93.87 | 93.99 |
| SVHN (OOD Detection) | AUROC | 97.44 | 97.44 | 97.92 | 97.95 | 97.42 | 97.51 |
| | AUPRC | 93.07 | 96.34 | 97.00 | 96.92 | 98.79 | 98.82 |
| Tiny-ImageNet (OOD Detection) | AUROC | 88.28 | 89.01 | 89.10 | 89.18 | 89.85 | 89.24 |
| | AUPRC | 88.13 | 89.81 | 89.76 | 89.72 | 89.18 | 89.26 |

spectively, an interesting option, in the presence of multiple datasets (e.g. acquired over time in a continual learning setting), could be applying the DRO loss component to different components of the training set, and assessing the results to robustly select $\delta$.

We also report the average EU estimation values of CreDEs-5 for each dataset in Table 5. Increasing the value of $\delta$ (i.e., giving less importance to the divergence between test and training

distributions) leads to a decreasing trend in the average EU estimates per dataset (particularly for ID CIFAR10 samples). This aligns with the intuition that, if the model is more uncertain about the divergence of the distributions (smaller $\delta$),

it should express a larger EU. Despite smaller uncertainty values at high $\delta$'s, the difference between ID and OOD samples remains noticeable. This explains why a $\delta$ closer to 1 does not always lead to low-performance OOD detection and why our model's OOD detection performance is robust against the choice of $\delta$.

Table 5: Averaged EU estimates of CreDEs-5 using various $\delta$.

| $\delta$ | 0.5 | 0.625 | 0.75 | 0.875 | 0.9375 | 0.96875 |
|---|---|---|---|---|---|---|
| CIFAR10 | 0.3557 | 0.0611 | 0.0586 | 0.0572 | 0.0222 | 0.0215 |
| SVHN | 1.6093 | 0.2450 | 0.2553 | 0.2593 | 0.1612 | 0.1574 |
| Tiny-ImageNet | 1.4996 | 0.2030 | 0.1957 | 0.1970 | 0.1025 | 0.1005 |

**Model Inference Complexity** Table 6 reports the parameter count and inference cost on one NVIDIA A100-SXM4-40GB GPU for a single SNN and CreNet on ImageNet. CreNets show a marginal increase in complexity due to its minor architectural modifications. More discussions on the inference and training complexity are presented in Appendix §C.

**Additional Experiments** Appendix §B.1 discusses the implementation and performance of EU quantification in CreDEs when using a different uncertainty measure for credal sets (the *generalized Hartley measure* [4, 36]). The results demonstrate that our CreDEs consistently enhance the quality of EU quantification, exhibiting robustness against different measures. Appendix §B.2 reports an ablation study for the hyperparameter $K$ of our PIDR Algorithm 2, which shows the effect of $K$ on CreDEs's uncertainty quantification and time cost. Appendix §B.3 assesses the ability of CreDEs to evaluate

Table 6: Model complexity of a ResNet50-based SNN and CreNet on ImageNet dataset.

| Model | Parameters (million) | Inference time per sample (ms) |
|---|---|---|
| SNN vs CreNet | 25.557 vs 27.606 | 5.5±0.2 vs 5.7±0.3 |

*total uncertainty* (TU) (as opposed to EU) in OOD detection, suggesting that our CreDEs also achieve an improved TU estimation, compared to DEs. Appendices §B.4, §B.5, and §B.6 compare the uncertainty quantification abilities of CreDEs versus those of traditional DEs that also apply the DRO strategy, DEs that apply the 'product of experts' [32] ensemble setting, several BNN baselines, respectively. CreDEs continue to demonstrate superior performance in uncertainty estimation. Appendix §B.7 assess CreDEs in a case study involving active learning [23, 54]. All these additional experiments demonstrate that our CreDEs deliver improved uncertainty quantification.

# 4 Conclusion

**Conclusion** In this paper, we introduced a novel *Credal-Set Neural Network* (CreNet) for classification tasks. Given any given input instance, CreNet is designed to predict a lower and an upper probability for each class, rather than a single probability value, thus providing an efficient and effective implementation of credal inference. We also proposed *Credal Deep Ensembles* (CreDEs), ensembles of CreNets, which extend the traditional deep ensemble idea to the credal domain. Extensive experimental validation was conducted on several OOD detection benchmarks, and across different network architectures and uncertainty measures. Compared to traditional Deep Ensembles, our CreDEs achieve higher test accuracy and lower ECE on ID samples, while significantly improving the quality of EU and TU estimation, leading in turn to strongly enhanced OOD detection performance. Hence, we believe our work can potentially improve neural network safety and reliability, and have wide applicability to real-world scenarios such as medical image analysis.

**Limitation** Despite the superior performance of CreDEs, neither traditional DEs nor CreDEs may be desirable when memory usage is stringent and computational resources are limited.

**Future Work** Three essential objectives of our future research include elaborating on statistical coverage guarantees of our CreDEs (outlined in Appendix §E.2), extending our framework to regression tasks (a roadmap is provided in Appendix §E.3), and assessing our CreDEs alongside other uncertainty-aware models in real-world applications comprehensively, like medical image analysis.

# Acknowledgement

We thank the anonymous reviewers for their valuable feedback. This work has received funding from the European Union's Horizon 2020 research and innovation program under grant agreement No. 964505 (E-pi).

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

# A  Mathematical Proof

**Toy Problem for Unavailability of Traditional SoftMax**   The traditional SoftMax activation function cannot be used to define the credal set, as it cannot ensure the condition in Eq. (2) when computing $[\boldsymbol{q}_L, \boldsymbol{q}_U]$ as $\boldsymbol{q}_L = \text{SoftMax}(\boldsymbol{a}_L)$ and $\boldsymbol{q}_U = \text{SoftMax}(\boldsymbol{a}_U)$, respectively.

For example, assuming that we have $\boldsymbol{a}_L := (-1, 0, 1)$ and $\boldsymbol{a}_U := (0, 1, 3)$ from CreNet, the $\boldsymbol{q}_L$ and $\boldsymbol{q}_U$ computed using SoftMax are:

$$\boldsymbol{q}_L = \text{SoftMax}(\boldsymbol{a}_L) = (0.0900, 0.2447, 0.6653), \;\; \boldsymbol{q}_U = \text{SoftMax}(\boldsymbol{a}_U) = (0.0420, 0.1142, 0.8438).$$

The resulting 'probability intervals' are not properly defined and appear unreasonable, as some lower bounds are considerably higher than the upper bounds.

**Mathematical Proof for Interval SoftMax**   The proof that Interval SoftMax in Eq. (4) does satisfy the conditions in Eq. (2) is straightforward:

$$
\begin{aligned}
\sum_{i=1}^{C} q_{L_i} &= \sum_{i=1}^{C} \frac{\exp(a_{L_i})}{\sum_{k \neq i}^{C}\exp(\frac{a_{U_k}+a_{L_k}}{2}) + \exp(a_{L_i})} \leq \sum_{i=1}^{C} \frac{\exp(\frac{a_{U_i}+a_{L_i}}{2})}{\sum_{k \neq i}^{C}\exp(\frac{a_{U_k}+a_{L_k}}{2}) + \exp(\frac{a_{U_i}+a_{L_i}}{2})} \\
&= 1 \leq \sum_{i=1}^{C} \frac{\exp(a_{U_i})}{\sum_{k \neq i}^{C}\exp(\frac{a_{U_k}+a_{L_k}}{2}) + \exp(a_{U_i})} = \sum_{i=1}^{C} q_{U_i}
\end{aligned}
\tag{16}
$$

# B  Additional Experiments

In this section, Appendix §B.1 discusses the implementation and performance of EU quantification of CreDEs using another uncertainty measure for credal sets: the generalized Hartley measure. Appendix §B.2 performs an ablation study on the value of the hyperparameter $K$ of the PIDR Algorithm 2. Appendix §B.3 assessed CreDEs' performance in the OOD detection task when quantifying uncertainty using TU instead of EU. Appendices §B.4, §B.5, and §B.6 compare CreDEs versus traditional DEs that also apply the DRO strategy, DEs that apply the 'product of experts' ensemble setting [32], and several BNN baselines, respectively. Appendix §B.7 assessed CreDEs in an active learning case study.

## B.1  Generalized Hartley Measure for EU Quantification of CreDEs

Uncertainty quantification in credal sets merits further investigation. For instance, recent research [35] has explored, e.g., the use of probability interval length as a measure of epistemic uncertainty, in the special case of binary classification. However, these measures cannot be readily extended to multi-class cases. Recently, the most established methods for decomposing the total uncertainty of credal sets are generalized Entropy [3, 36] and the generalized Hartley Measure [4, 36].

**Definition**   The generalized Hartley measure [4], $\text{GH}(\mathbb{Q})$, measures the non-specificity across the distributions in the credal set, and can be seen as a proxy for its volume [36]. Mathematically, $\text{GH}(\mathbb{Q})$ calculates the expectation of the Hartley measure [26] over all possible subsets $\mathbb{B}$ on the target space $\mathbb{Y}$,[2] as follows [4]:

$$\text{GH}(\mathbb{Q}) = \sum_{\mathbb{B} \subseteq \mathbb{Y}} \text{m}_{\mathbb{Q}}(\mathbb{B}) \cdot \log_2(|\mathbb{B}|), \tag{17}$$

in which $\text{m}_{\mathbb{Q}}$ denotes the mass assignment function associated to $\mathbb{Q}$ and $|\mathbb{B}|$ indicates the cardinality of $\mathbb{B}$. $\text{m}_{\mathbb{Q}}(\mathbb{B})$ can be computed using the Möbius inverse of the capacity function $\nu_{\mathbb{Q}}$ [36], as follows:

$$\text{m}_{\mathbb{Q}}(\mathbb{B}) = \sum_{\mathbb{A} \subseteq \mathbb{B}} (-1)^{|\mathbb{B} \backslash \mathbb{A}|} \nu_{\mathbb{Q}}(\mathbb{A}), \tag{18}$$

where $\mathbb{B} \backslash \mathbb{A} = \{y | y \in \mathbb{B} \text{ and } y \notin \mathbb{A}\}$ and $\nu_{\mathbb{Q}}$ describes the lower probability of all possible subsets $\mathbb{A} \subseteq \mathbb{B}$.

**Efficient Implementation in CreDEs**   One of the reasons that hinder the application of the Generalized Hartley Measure is its computational complexity. In our work, we proposed an efficient implementation and an approximate approach (the PIDR Algorithm 2) for computing it.

---

[2]In classification, target space $\mathbb{Y}$ comprises a finite set of class labels, namely $\mathbb{Y} = \{y_1, ..., y_C\}$.

In our case, the lower probability $\nu_{\mathbb{Q}}(\mathbb{A})$ associated with the predicted credal set can be readily computed as follows:

$$\nu_{\mathbb{Q}}(\mathbb{A}) = \max\left(\sum_{y_j \in \mathbb{A}} q^*_{L_j}, 1 - \sum_{y_j \notin \mathbb{A}} q^*_{U_j}\right), \tag{19}$$

where $q^*_L$ and $q^*_U$ are the reachable lower and upper probability values per class in the defined credal set. They can be easily obtained from Eq. (12). Figure 3 illustrates how the lower and upper probabilities $q_{L_i}, q_{U_i}$ that determine the credal set $\mathbb{Q}$ may differ from the probabilities $q^*_{L_i}$ and $q^*_{U_i}$ *actually reachable* for each class within $\mathbb{Q}$.

The full $\text{GH}(\mathbb{Q})$ calculation process is presented in Algorithm 3.

---

**Algorithm 3** $\text{GH}(\mathbb{Q})$ Calculation

---

   **Input:** $[\boldsymbol{q}^*_L, \boldsymbol{q}^*_U] := \{[q^*_{L_i}, q^*_{U_i}]\}_{i=1}^C$; Target space $\mathbb{Y}$
   **Output:** $\text{GH}(\mathbb{Q})$
   Initialize: $\text{GH}(\mathbb{Q}) = 0$
   **for all** $\mathbb{B} \subseteq \mathbb{Y}$ **and** $|\mathbb{B}| \geq 2$ **do**
      Initialize: $\text{m}_{\mathbb{Q}}(\mathbb{B}) = 0$
      **for all** $\mathbb{A} \subseteq \mathbb{B}$ **do**
         Compute $\nu_{\mathbb{Q}}(\mathbb{A})$ using Eq. (19)
         $\text{m}_{\mathbb{Q}}(\mathbb{B}) = \text{m}_{\mathbb{Q}}(\mathbb{B}) + (-1)^{|\mathbb{B} \setminus \mathbb{A}|} \cdot \nu_{\mathbb{Q}}(\mathbb{A})$ (Eq. (18))
      **end for**
      $\text{GH}(\mathbb{Q}) = \text{GH}(\mathbb{Q}) + \text{m}_{\mathbb{Q}} \cdot \log_2(|\mathbb{B}|)$ (Eq. (17))
   **end for**

---

Although the use of probability intervals simplifies the calculation of $\text{GH}(\mathbb{Q})$ in general, a significant challenge arises for large values of $C$ (e.g. $C = 100$) due to the complexity of involving subsets of $C$.

However, when applying our proposed PIDR Algorithm 2, the dimension of the probability interval is reduced from $C$ to $K$; therefore, calculating $\text{GH}(\mathbb{Q})$ requires only $2^K$ subsets.

**Experimental Validation of GH$(\mathbb{Q})$ for OOD Detection** The OOD detection results for CreDEs-5 using $\text{GH}(\mathbb{Q})$ are shown in Tables 7, 8 and in Figure 9. Probability Interval Dimension Reduction (PIDR) (Algorithm 2) is utilized with settings $K = 4$ and $K = 10$ when computing $\text{GH}(\mathbb{Q})$ for dataset pairs containing CIFAR10 and CIFAR100/ImageNet, respectively. The results verify that

- Our CreDEs consistently enhance the quality of EU quantification, exhibiting robustness across different uncertainty measures, i.e., the generalized Shannon entropy and the generalized Hartley measure. This improved EU quantification leads to better OOD detection performance compared to Deep Ensemble baselines.

- The proposed PIDR algorithm ensures an efficient implementation of the generalized Hartley measure in our framework. An ablation study on PIDR's hyperparameter is conducted in Appendix §B.2.

Note, however, that applying $K = 10$ for the setting ImageNet vs ImageNet-O does not yield a better result, due to the coarseness of approximating 1000 classes using only 10. This suggests that computing $\text{GH}(\mathbb{Q})$ is still challenging for tasks involving 1000 or more classes.

## B.2 Ablation Study on Hyperparameter of PIDR Algorithm

**Effect on GH$(\mathbb{Q})$ Quantification** Figure 10 illustrates the influence of various settings of $K$ on $\text{GH}(\mathbb{Q})$ quantification. The average value $\text{GH}(\mathbb{Q})$ suggests that the use of the PIDR Algorithm 2) results in an underestimated $\text{GH}(\mathbb{Q})$ value, compared to the result without using PIDR ($K = 10$). Consequently, increasing the value of $K$ enhances OOD detection performance. However, as $K$ grows execution time increases exponentially, due to the iterative calculations of $\text{m}_{\mathbb{Q}}(\mathbb{B})$ and $\nu_{\mathbb{Q}}(\mathbb{A})$ in Algorithm 3 across $2^K$ subsets. The time cost is measured on a single Intel Xeon Gold 8358 CPU@2.6 GHz. While higher than the time cost for EU calculation of DEs ($4.1e^{-4}$ ms), this figure shows that calculating GH for 10 classes ($K = 10$ for 17 ms) remains practical without actual computational constraints. Besides, the numbers reported are for GH calculation without any optimization: a more

Table 7: OOD detection AUROC and AUPRC performance (%, ↑) between CreDEs-5 and DEs-5 based on ResNet50 using EU as uncertainty metrics on CIFAR10/100 vs. SVHN/Tiny-ImageNet and ImageNet vs. ImageNet-O. Results are averaged over 15 runs. Best results are in bold.

| ID Samples | | CIFAR10 | | | | CIFAR100 | | | | ImageNet | |
|---|---|---|---|---|---|---|---|---|---|---|---|
| OOD Samples | | SVHN | | Tiny-ImageNet | | SVHN | | Tiny-ImageNet | | ImageNet-O | |
| Performance Indicator | | AUROC | AUPRC | AUROC | AUPRC | AUROC | AUPRC | AUROC | AUPRC | AUROC | AUPRC |
| DEs-5 | $H(\bar{q})-\tilde{H}(q)$ | 89.58±0.93 | 92.29±1.00 | 86.87±0.20 | 83.02±0.16 | 73.83±1.97 | 84.96±1.25 | 78.80±0.20 | 74.68±0.27 | 65.03±0.53 | 62.77±0.38 |
| CreDEs-5 | $\overline{H}(\mathbb{Q})-\underline{H}(\mathbb{Q})$ | **96.55±0.25** | **98.17±0.17** | 88.10±0.26 | 87.85±0.35 | 78.55±1.15 | 86.57±0.65 | 82.54±0.26 | 77.60±0.44 | **67.82±0.06** | **62.80±0.12** |
| (ours) | GH($\mathbb{Q}$) | **96.72±0.24** | **98.25±0.17** | **89.54±0.16** | **88.74±0.24** | **79.23±1.19** | **87.17±0.66** | **83.01±0.24** | **78.95±0.44** | 63.46±0.06 | 58.13±0.10 |

Table 8: AUROC and AUPRC scores (%, ↑) for OOD detection on CIFAR10 vs SVHN/Tiny-ImageNet. Results averaged over 15 runs. The Best results are in bold.

| | | | CIFAR10 vs SVHN | | CIFAR10 vs Tiny-ImageNet | |
|---|---|---|---|---|---|---|
| | | | AUROC | AUPRC | AUROC | AUPRC |
| | DEs-5 | $H(\bar{q})-\tilde{H}(q)$ | 82.19±0.82 | 87.52±0.81 | 78.58±0.15 | 73.28±0.23 |
| VGG16 | CreDEs-5 | $\overline{H}(\mathbb{Q})-\underline{H}(\mathbb{Q})$ | **87.68±0.73** | **93.47±0.57** | **82.56±0.28** | 80.81±0.52 |
| | (Ours) | GH($\mathbb{Q}$) | 86.99±0.72 | 93.18±0.41 | 82.23±0.18 | **80.83±0.24** |
| | DEs-5 | $H(\bar{q})-\tilde{H}(q)$ | 77.71±1.67 | 88.73±0.32 | 82.27±0.79 | 78.85±0.81 |
| ViT Base | CreDEs-5 | $\overline{H}(\mathbb{Q})-\underline{H}(\mathbb{Q})$ | 88.57±2.08 | 93.24±1.25 | 88.73±0.32 | 87.84±0.52 |
| | (Ours) | GH($\mathbb{Q}$) | **89.07±1.66** | **93.32±1.06** | **89.19±0.42** | **88.21±0.58** |

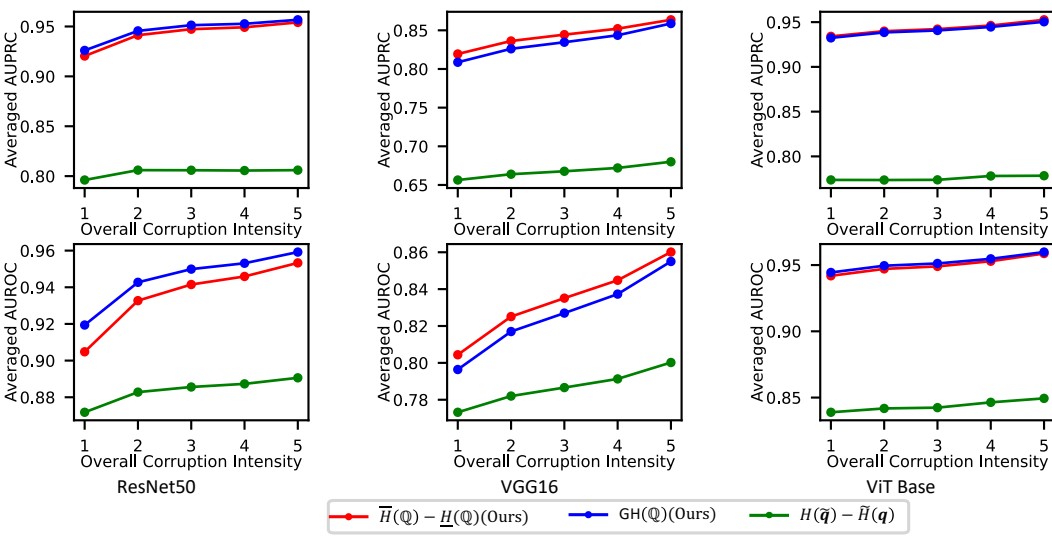

Figure 9: OOD detection on CIFAR10 vs CIFAR10-C against increased corruption intensity, using ResNet50, VGG16, and ViT Base as backbones.

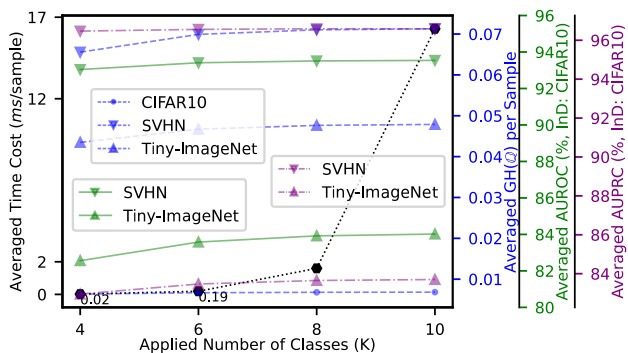

Figure 10: Average time cost of GH($\mathbb{Q}$) (black dotted line) and GH($\mathbb{Q}$) value per sample across various datasets (blue lines), along with the AUROC/AUPRC scores (green/purple lines) for OOD detection versus increasing values of $K$.

efficient code implementation could significantly reduce the cost. The effect of various settings of $K$ on $\overline{H}(\mathbb{Q})$ is shown as follows.

**Effect on TU Quantification** In this experiment, we examine the effect of $K$ on TU estimation. Figure 11 shows the average values of TU estimates ($\overline{H}(\mathbb{Q})$) sample, together with the AUROC and AUPRC scores for CIFAR100 vs. SVHN/Tiny-ImageNet. The results indicate that applying PIDR (Algorithm 2) tends to underestimate TU values. Consequently, increasing the value of $K$ improves the OOD detection performance, but it also leads to an increase in execution time. This is because solving the constrained optimization problem in Eq. (13) involves more variables and constraints.

The reported time cost is measured on a single Intel Xeon Gold 8358 CPU@2.6 GHz, without optimization in the calculation process. We believe a more efficient code implementation could significantly mitigate this.

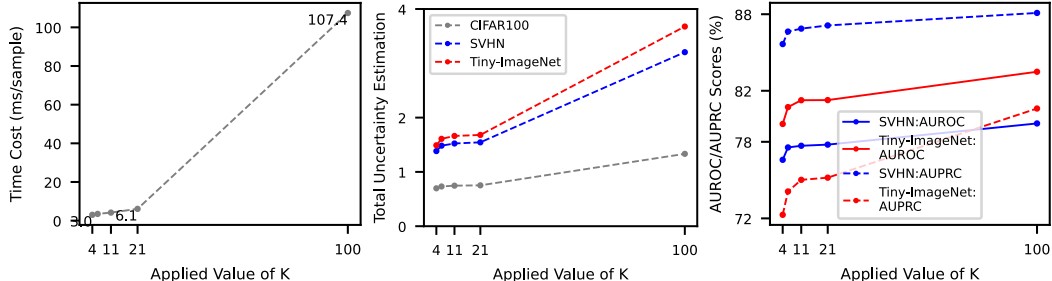

Figure 11: Average $\overline{H}(\mathbb{Q})$ time cost, average $\overline{H}(\mathbb{Q})$ value per sample, and OOD performance on the OOD detection benchmark (CIFAR100 vs. SVHN/Tiny-ImageNet) for increasing values of $K$.

### B.3 Total Uncertainty Estimation Evaluation via OOD Detection

In further additional experiments, we also assess the quality of the *total uncertainty* (TU) estimates produced by CreDEs-5 on the various OOD detection benchmarks [43, 54]. The results in Tables 9 and 10 consistently demonstrate CreDEs' improved OOD detection performance using TU as a metric.

### B.4 Comparison between CreDEs and DEs with DRO Strategy

In this experiment, we additionally train 15 ResNet50-based SNNs on the CIFAR10 and CIFAR100 datasets using the DRO loss component and the same training strategy as in Algorithm 1, respectively. For a fair comparison, we set $\delta = 0.5$ (just as for CreNets), using the same random seeds and training epochs. Other training configurations are described in Appendix §C. We named the resulting deep ensembles with 5 ensemble members as DEs*-5.

Table 9: OOD detection AUROC and AUPRC performance (%, ↑) between CreDEs-5 and DEs-5 based on ResNet50 using TU as uncertainty metrics on CIFAR10/100 vs. SVHN/Tiny-ImageNet and ImageNet vs. ImageNet-O. Results are averaged over 15 runs. Best results in bold.

| ID Samples | | CIFAR10 | | | | CIFAR100 | | | | ImageNet | |
|---|---|---|---|---|---|---|---|---|---|---|---|
| OOD Samples | | SVHN | | Tiny-ImageNet | | SVHN | | Tiny-ImageNet | | ImageNet-O | |
| Performance Indicator | | AUROC | AUPRC | AUROC | AUPRC | AUROC | AUPRC | AUROC | AUPRC | AUROC | AUPRC |
| DEs-5 | $H(\bar{q})$ | 94.80±0.43 | 97.26±0.29 | 88.80±0.19 | 87.21±0.29 | 78.53±1.94 | **88.83±1.01** | 80.75±0.15 | 77.65±0.19 | 50.20±0.07 | 50.43±0.06 |
| CreDEs-5 | $\overline{H}(\mathbb{Q})$ | **95.71±0.42** | **97.73±0.27** | **89.02±0.10** | **88.02±0.15** | **79.44±1.45** | 88.10±0.79 | **83.49±0.17** | **80.61±0.33** | **67.56±0.06** | **62.79±0.18** |

Table 10: OOD detection AUROC and AUPRC performance (%, ↑) between CreDEs-5 and DEs-5 based on VGG16 and ViT Base using TU as uncertainty metrics on CIFAR10 vs. SVHN/Tiny-ImageNet. Results are averaged over 15 runs. Best results in bold.

| | | VGG16 | | | | ViT Base | | | |
|---|---|---|---|---|---|---|---|---|---|
| | | SVHN (OOD) | | Tiny-ImageNet (OOD) | | SVHN (OOD) | | Tiny-ImageNet (OOD) | |
| | | AUROC | AUPRC | AUROC | AUPRC | AUROC | AUPRC | AUROC | AUPRC |
| DEs-5 | $H(\bar{q})$ | 84.50±0.49 | 90.78±0.35 | 79.40±0.10 | 75.91±0.14 | 79.80±1.75 | 87.97±1.17 | 83.81±0.81 | 81.68±0.89 |
| CreDEs-5 | $\overline{H}(\mathbb{Q})$ | **87.05±0.80** | **93.36±0.42** | **82.14±0.14** | **80.81±0.16** | **87.30±1.77** | **92.24±1.15** | **88.17±0.44** | **86.94±0.60** |

We compare test accuracy and ECE for DEs*-5 and CreDEs-5 in Table 11, and their OOD detection performance on the CIFAR10/100 (ID) vs SVHN/Tiny-ImageNet (OOD) benchmark in Table 12.

Table 11: Test accuracy and ECE of DEs*-5 and CreDEs-5 on the CIFAR10 and CIFAR100 datasets. Best results in bold.

| | | CIFAR10 | | CIFAR100 | |
|---|---|---|---|---|---|
| | | Test Accuracy | ECE | Test Accuracy | ECE |
| DEs*-5 | | 91.53±0.22 | 0.0159±0.0019 | 68.34±0.52 | 0.0372±0.0033 |
| CreDEs-5 | $\hat{i}_{min}$ | **93.75±0.11** | **0.0092±0.0016** | **79.54±0.21** | 0.0366±0.0025 |
| | $\hat{i}_{max}$ | 93.74±0.11 | 0.0108±0.0017 | 79.65±0.19 | **0.0268±0.0023** |

Table 12: OOD detection performance comparison of DEs*-5 and CreDEs-5 using the dataset pairs CIFAR10/100 (ID) vs SVHN/Tiny-ImageNet (OOD).

| | | CIFAR10 (ID) | | | | CIFAR100 (ID) | | | |
|---|---|---|---|---|---|---|---|---|---|
| | | SVHN (OOD) | | Tiny-ImageNet (OOD) | | SVHN (OOD) | | Tiny-ImageNet (OOD) | |
| | | AUROC | AUPRC | AUROC | AUPRC | AUROC | AUPRC | AUROC | AUPRC |
| TU | DEs*-5:$H(\bar{q})$ | 91.82±0.96 | 95.13±0.70 | 86.26±0.30 | 84.09±0.42 | 78.70±1.61 | **88.20±0.91** | 76.99±0.28 | 73.03±0.37 |
| | CreDEs-5:$\overline{H}(\mathbb{Q})$ | **95.71±0.42** | **97.73±0.27** | **89.02±0.10** | **88.02±0.15** | **79.44±1.45** | 88.10±0.17 | **83.49±0.17** | **80.61±0.33** |
| EU | DEs*-5:$H(\bar{q})-\tilde{H}(q)$ | 87.21±1.49 | 91.09±1.39 | 84.58±0.30 | 80.80±0.42 | 74.38±1.39 | 84.67±0.86 | 75.27±0.38 | 70.80±0.48 |
| | CreDEs-5:$\overline{H}(\mathbb{Q})-\underline{H}(\mathbb{Q})$ | 96.55±0.25 | 98.17±0.17 | 88.10±0.26 | 87.85±0.35 | 78.55±1.15 | 86.57±0.65 | 82.54±0.26 | 77.60±0.44 |
| | CreDEs-5:$GH(\mathbb{Q})$ | **96.72±0.24** | **98.25±0.17** | **89.54±0.16** | **88.74±0.24** | **79.23±1.19** | **87.17±0.66** | **83.01±0.24** | **78.95±0.44** |

The reported results demonstrate that CreDEs-5 outperforms DEs*-5 ensembles by achieving higher test accuracy and lower ECE values. Concerning OOD detection tasks, it can be found that CreDEs in general improve the AUPRC and AUROC scores using either the TU or the EU metric, pretty much across the board. These results suggest that CreDEs provide higher-quality EU and TU estimation.

In Table 12, a 0.1% drop in AUPRC using the TU metric can be observed. However, remember that CreDEs calculate TU (using the upper entropy) by solving a constrained optimization problem in Eq. (13) using a numerical solver from SciPy. The slight performance decrease is likely due to numerical errors during the optimization process.

## B.5 Comparison between CreDEs and DEs with POE Ensemble Setting

In this experiment, we conduct a comparison between our CreDEs and DEs with the 'product of experts' (POE) [32] ensemble setting, as opposed to the more commonly employed 'mixture of

experts' approach in our primary analysis. Here, DEs$^p$-5 denotes the deep ensembles that process the final predictions from the ensemble members using the POE strategy. The experimental setup mirrors that used by OOD detection benchmarks, involving data pairs CIFAR10 (ID) vs SVHN/Tiny-ImageNet (OOD).

Table 13 shows that DEs$^p$-5 could improve test accuracy but significantly reduce the calibration performance of DEs-5 (larger ECE values). Among these comparisons, CreDEs-5 emerged as the most superior method. Furthermore, we evaluate the uncertainty estimation through the OOD detection benchmark. Specifically, =the entropy of the final prediction of DEs$^p$-5 is calculated to quantify the total uncertainty. For CreDEs-5 and DEs-5, we use the upper entropy, $\overline{H}(\mathbb{Q})$, and $H(\tilde{q})$, respectively. The results in Table 14 consistently demonstrate the superior performance of our method. Although the POE strategy improves the test accuracy of classical DEs, it significantly degrades calibration performance and leads to inferior OOD detection performance.

Table 13: Test accuracy (ACC) (%) and ECE comparison on the CIFAR10 dataset, using the ResNet50, VGG16, and ViT Base architectures.

| | ResNet50 | | VGG16 | | ViT Base | |
|---|---|---|---|---|---|---|
| | ACC | ECE | ACC | ECE | ACC | ECE |
| CreDEs-5 ($\hat{i}_{\max}$) | **93.74±0.11** | **0.0109±0.0017** | **87.92±0.11** | **0.0611±0.0012** | **93.59±0.39** | **0.0104±0.0012** |
| CreDEs-5 ($\hat{i}_{\min}$) | **93.75±0.11** | **0.0092±0.0016** | **87.94±0.11** | **0.0203±0.0014** | **93.60±0.40** | **0.0107±0.0014** |
| DEs-5 | 93.32±0.13 | 0.0131±0.0010 | 85.53±0.10 | 0.0815±0.0011 | 90.43±0.97 | 0.0181±0.0019 |
| DEs$^p$-5 | 93.47±0.11 | 0.0610±0.0011 | 85.55±0.08 | 0.1368±0.0008 | 90.56±0.90 | 0.0894±0.0087 |

Table 14: OOD detection performance comparison (%) on CIFAR10 vs SVHN/Tiny-ImageNet, using the ResNet50, VGG16, and ViT Base architectures.

| | ResNet50 | | | | VGG16 | | | | ViT Base | | | |
|---|---|---|---|---|---|---|---|---|---|---|---|---|
| | SVHN | | Tiny-ImageNet | | SVHN | | Tiny-ImageNet | | SVHN | | Tiny-ImageNet | |
| | AUROC | AUPRC | AUROC | AUPRC | AUROC | AUPRC | AUROC | AUPRC | AUROC | AUPRC | AUROC | AUPRC |
| CreDEs-5 | **95.71±0.42** | **97.73±0.27** | **89.02±0.10** | **88.02±0.15** | **87.05±0.80** | **93.36±0.42** | **82.14±0.14** | **80.81±0.16** | **87.30±1.77** | **92.24±1.15** | **88.17±0.44** | **86.94±0.60** |
| DEs-5 | 94.80±0.43 | 97.26±0.29 | 88.80±0.19 | 87.21±0.29 | 84.50±0.49 | 90.78±0.35 | 79.40±0.10 | 75.91±0.14 | 79.80±1.75 | 87.97±1.17 | 83.81±0.81 | 81.67±0.89 |
| DEs$^p$-5 | 93.90±0.24 | 96.10±0.21 | 88.03±0.20 | 84.11±0.32 | 84.10±0.22 | 89.83±0.16 | 78.11±0.08 | 72.23±0.16 | 82.41±1.56 | 88.51±0.95 | 83.21±1.02 | 78.24±1.17 |

## B.6 Comparison between CreDEs and Bayesian Neural Networks

As discussed in the main body, the main reason for excluding Bayesian neural network (BNN) approaches in our main evaluation is that they generally have difficulty scaling to large datasets and complex network architectures [54]. In this section, we conducted an additional comparison between CreDEs and DEs, MCDropout [22], and two TensorFlow-standardized BNNs (BNN-R [51] and BNN-F [76]). All the models are trained on the ResNet50 for the CIFAR10 dataset from scratch. The input data shape is (32, 32, 3). The Adam optimizer is applied with a learning rate scheduler, initialized at 0.001. The learning rate is subject to a reduction of 0.1 at epochs 80 and 120. For BNNs, 10 forward passes are used for uncertainty estimation.

The uncertainty evaluation via OOD detection on the CIFAR10 vs SVHN/Tiny-ImageNet dataset is reported in Table 15. The results consistently demonstrate the significant improvements of our CreDEs.

## B.7 Case Study on Active Learning Settings

Active learning (AL) aims to efficiently train models with minimal data by acquiring additional samples from a vast pool of unlabeled data, which are then labeled by experts [11]. After each acquisition step, the model is retrained using the expanded training set. The iterative process continues until either the desired accuracy or the maximum allowable acquired samples are reached. Efficient data acquisition can be a reliable estimate of the uncertainty of models [23, 54].

**Setup** We deploy CreDEs-5 ($\delta = 0.5$) and DEs-5 (baseline) using the ResNet18 architecture and utilizing clean MNIST samples in the pool set. TU and EU estimations from each approach for the acquisition functions are utilized. We begin with an initial training set of 20 randomly selected MNIST points.

Table 15: OOD detection AUROC and AUPRC performance (%, ↑) between CreDEs-5 and Bayesian models based on ResNet50 using EU and TU as uncertainty metrics on CIFAR10 vs. SVHN/Tiny-ImageNet. Results are averaged over 15 runs. The best results are in bold. The 'drop' denotes the dropout rate applied to MCDropout.

| Model | | Epistemic Uncertainty Measure as Metric | | | | Total Uncertainty Measure as Metric | | | |
|---|---|---|---|---|---|---|---|---|---|
| | | SVHN (OOD) | | Tiny-ImageNet (OOD) | | SVHN (OOD) | | Tiny-ImageNet (OOD) | |
| | | AUROC | AUPRC | AUROC | AUPRC | AUROC | AUPRC | AUROC | AUPRC |
| CreDEs-5 | | **79.14±1.49** | **86.84±1.18** | **82.85±0.29** | **80.71±0.42** | **81.00±0.75** | **88.66±0.46** | **84.06±0.11** | **82.16±0.13** |
| DEs-5 | | 73.53±1.65 | 83.81±1.42 | 76.13±0.58 | 70.86±0.67 | 77.93±0.65 | 84.92±0.39 | 80.22±0.26 | 76.94±0.30 |
| BNN-R | | 70.30±3.55 | 82.41±2.45 | 72.91±2.01 | 67.82±2.10 | 73.37±2.00 | 82.69±1.58 | 73.98±1.85 | 70.52±1.89 |
| BNN-F | | 70.15±4.38 | 82.04±3.01 | 73.66±1.46 | 68.52±1.53 | 73.77±2.62 | 82.90±1.71 | 74.57±1.30 | 71.11±1.29 |
| MCDropout | 0.1 drop | 74.19±1.55 | 82.93±1.01 | 75.04±0.77 | 68.25±1.31 | 76.92±1.85 | 85.93±1.22 | 77.48±0.56 | 73.63±0.62 |
| | 0.4 drop | 61.66±1.89 | 73.47±1.27 | 67.24±1.36 | 59.55±1.41 | 79.25±0.96 | 86.04±0.77 | 76.04±0.57 | 72.73±0.65 |

In each iteration, we acquire the 5 samples with the highest reported uncertainty estimates (EU or TU per model). After each step, we train models using the Adam optimizer for 20 epochs and select the one with the best accuracy from the validation set. AL process stops when the training set size reaches 150.

**Results** Figure 12 shows the result comparison between CreDEs-5 and DEs-5 using TU and EU estimates as the acquisition functions per model. In the evaluation using MNIST, aiming for a 90% accuracy or a maximum sample count of 150, CreDEs-5 employing acquisition functions TU ($\overline{H}(\mathbb{Q})$) and EU (GH($\mathbb{Q}$)), demonstrates superior performance compared to DEs-5 using TU ($H(\tilde{q})$). In addition, CreDEs-5 with EU ($\overline{H}(\mathbb{Q}) - \underline{H}(\mathbb{Q})$) outperforms DEs-5 with EU ($H(\tilde{q}) - \tilde{H}(q)$). The additional evidence verifies the improved quality of EU and TU estimation of CreDEs, compared to DEs. In future work, we aim to explore the potential integration of our methods into other active learning benchmarks [70] and real-world applications or further improve on them.

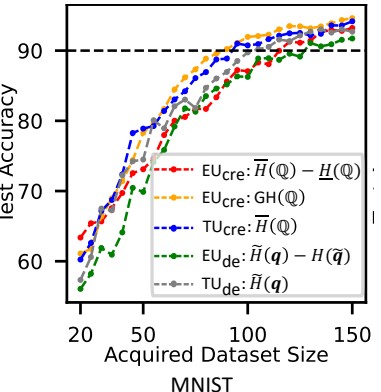

Figure 12: AL experiments using different acquisition functions. Achieved test accuracy vs. acquired training set size.

## C Experiment Implementation Details

For the main experiments on the ResNet50 backbone, we used two Tesla P100-SXM2-16GB GPUs as devices to independently train 15 SNNs and CreNets using CIFAR10 and CIFAR100 datasets. The input shape of both networks was (224, 224, 3). We employed the Adam optimizer, with a learning rate scheduler set at 0.001 and reduced to 0.0001 during the last five training epochs.

Figure 13 shows the averaged training and validation accuracy for training process monitoring.

In the ImageNet experiments, we employed three NVIDIA A100-SXM4-80GB GPUs. To create deep ensembles, we independently retrained 15 deep SNNs based on a pre-trained ResNet50 model for 3 epochs, using the Adam optimizer with an initialized learning rate of $1e^{-6}$. For CreDEs, we initialized CreNet weights using a pre-trained ResNet50 model and independently retrained 15 CreNet models for 5 epochs, using the Adam optimizer with an initialized learning rate of $1e^{-5}$. The choice of a larger learning rate value and epoch count for CreNets is a consequence of their modified final later compared to SNNs. For the ablation study on various network architectures, we again utilized two Tesla P100-SXM2-16GB GPUs and one NVIDIA A100-SXM4-80GB GPU as devices to independently train 15 SNNs and CreNets, based on VGG16 and ViT Base architectures, respectively, and using the CIFAR10 dataset. VGG16-based SNNs and CreNets were trained for 20 epochs. SNNs and CreNets using the ViT Base backbone were trained for 25 and 40 epochs, respectively. The input

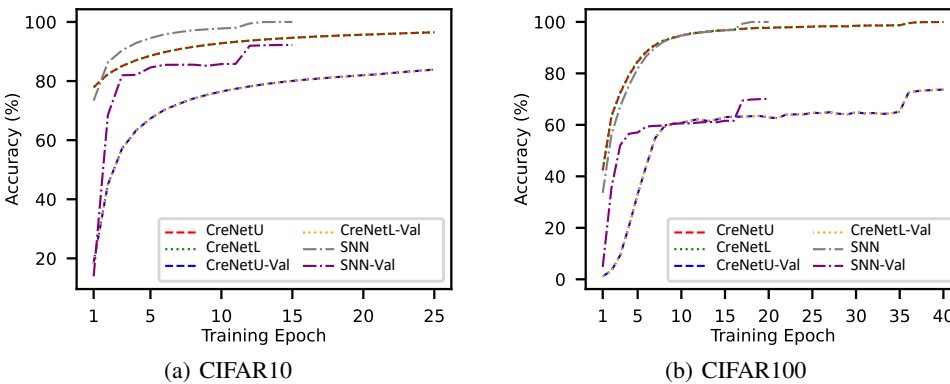

(a) CIFAR10             (b) CIFAR100

Figure 13: Averaged training and validation (Val) accuracy (%) for CreNets and SNNs over 15 runs. The U and L in the labels of CreNets represent accuracies associated with upper and lower probability bounds, namely $\hat{i}_{\max}$ and $\hat{i}_{\min}$, respectively.

shape for both networks was set to (224, 224, 3). For optimization, we employed the Adam optimizer with a learning rate scheduler initialized at 0.001 and reduced to 0.0001 during the final 5 training epochs.

**Training Complexity** We did not include the report of training time complexity in the main paper as CreNets use a custom training loop, unlike the TensorFlow-standardized training of standard neural networks (SNNs), precluding a fair comparison.

Nevertheless, we did train a single CreNet and a single SNN based on the ResNet50 architecture on the CIFAR10 dataset, from scratch and on a single A100 GPU. The training time per epoch is 16.36s for the SNN and 73.77s for CreNet, respectively. Given the evidence that CreNets only marginally increases the inference time (single forward pass), we are optimistic that by standardizing and optimizing the customized training loop and adopting a more efficient code implementation of Algorithm 1, we could significantly reduce the training load.

**Further Discussion on Inference Complexity** As we discussed previously, regarding inference time, doubling the final layer nodes would slightly increase the inference time. For instance, the inference time per sample for a ResNet50 architecture on the ImageNet dataset is 5.5 ms for a single standard neural network, vs 5.7 ms for a single CreNet (a marginal increase). The inference cost on the CIFAR10/100 dataset reported in Table 16 further demonstrates a slight increase in inference complexity in our method. Moreover, Table 17 presents the inference cost, evaluated on a single AMD EPYC 7643 48-core CPU. The results indicate no significant overhead of our CreDEs and also demonstrate that employing VGG16, a lighter model architecture compared to ResNet50, substantially reduces the inference cost for both DEs and CreDEs.

Table 16: Complexity comparison between ResNet50-based SNNs and CreNets using CIFAR10/100 datasets. The inference cost per dataset is measured by a single NVIDIA P100 SXM2-16GB GPU for both models.

| Dataset | Model | Parameters (million) | Inference time per sample (ms) |
|---------|-------|---------------------|-------------------------------|
| CIFAR10 | SNNs vs CreNets | 26.216 vs 26.221 | 60.6±0.7 vs 63.0±1.1 |
| CIFAR100 | SNNs vs CreNets | 26.262 vs 26.314 | 62.5±0.5 vs 63.1±0.7 |

Table 17: Inference cost comparison on CPU between SNNs and CreNets per single CIFAR10 input of different architectures.

| | VGG16 (ms) | ResNet50 (ms) |
|---|---|---|
| SNNs vs CreNets | 19.2±3.8 vs 23.1±5.2 | 148.2±49.0 vs 163.3±39.4 |

Regarding the uncertainty estimation cost, we report the cost of calculating the Generalized Hartley (GH) measure and the upper entropy in Figures 10 and 11, respectively. For example, the time cost for GH calculation for CIFAR10 without approximation is 17 ms (0.02 ms in the reduced case considering 4 out of 10 classes) while calculating the EU in deep ensembles for CIFAR10 takes

$1 \times 10^{-4}$ms, measured on the same single CPU. Though higher CreDEs remain practical without actual computational constraints. In addition, the reported numbers are without code efficiency optimization: a more efficient code implementation could significantly reduce the cost.

The practical takeaway here is that, as demonstrated by extensive experimental variation, our CreDEs exhibit strong potential to enhance the uncertainty quantification performance of DEs in real-world applications, with only a modest increase in computational complexity. However, if DEs are already deemed impractical due to computational limitations, our CreDEs would not be a suitable alternative.

## D Discussions on Ensemble Approaches

### D.1 Rationale for Averaging Ensemble Strategy

The randomness of parameter initialization in neural networks is one of the reasons leading to (epistemic) uncertainty about the 'ground-truth' model. As we gather more information, both epistemic and total uncertainty should decrease. For example, if we assume that we can train an infinite number of standard neural networks, then the Deep Ensembles would eliminate the source of ignorance caused by the randomness of parameter initialization.

Our proposed averaging approach to creating an ensemble of CreNets follows a similar rationale. Specifically, if we aggregate an infinite number of ensemble members, the uncertainty caused by the randomness would vanish. The outputted probability interval of CreDEs, primarily acknowledges the lack of precise insights into the divergence between the training and test distributions.

### D.2 Possible Alternative Ensemble Approaches

CreDEs aggregate predictions from multiple individually trained CreNets, producing credal sets based on probability intervals. In addition to averaging, two alternative approaches, namely union (disjunctive combination) and intersection (conjunctive combination) [17], can be envisaged.

These alternative methods are illustrated in Figure 14.

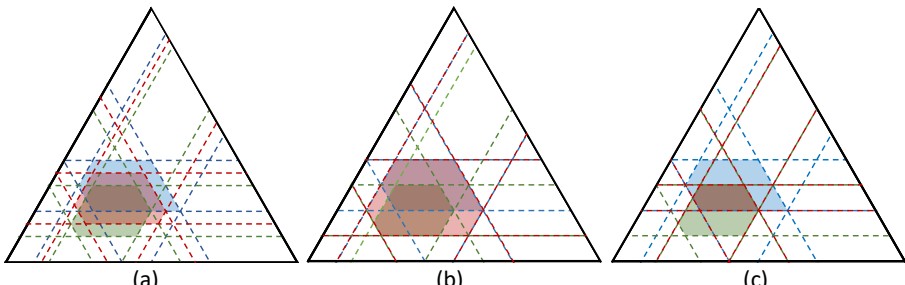

Figure 14: Representation of three ensemble approaches: averaging (a), union (b), and intersection (c). In each subfigure, the ultimate credal set (highlighted in dark red) is formed by aggregating two individual credal sets, each constrained by probability intervals indicated in light green and blue, respectively.

**Averaging** CreDEs average the upper and lower probabilities per class from $M$ individually-trained CreNets and predict the aggregated probability intervals, denoted as $[\tilde{q}_L^*, \tilde{q}_U^*]$, as discussed in Eq. (14). It can be proved that $[\tilde{q}_L^*, \tilde{q}_U^*]$ is guaranteed to generate a non-empty credal set, as follows:

$$\sum_{i=1}^{C} \tilde{q}_{L_i}^* = \frac{1}{M} \sum_{m=1}^{M} \sum_{i=1}^{C} q_{L_{m_i}}^* \leq 1 \leq \frac{1}{M} \sum_{m=1}^{M} \sum_{i=1}^{C} q_{U_{m_i}}^* = \sum_{i=1}^{C} \tilde{q}_{U_i}^*. \tag{20}$$

The semantic behind averaging is that we equally trust all pieces of information (individual credal sets) without judging the authenticity of the information. Similar to traditional deep ensembles (DEs), the averaging ensemble approach can alleviate the influence of training process randomness.

**Union** Given a collection of convex probability intervals, denoted as $\{[q_{L_m}^*, q_{U_m}^*]\}_{m=1}^{M}$, De Campos et al. have proposed the computationally efficient way to calculating the union of credal sets [17], as

follows:

$$\tilde{q}^*_{L_i} = \min_{m \in \{1,...,M\}} q^*_{L_{m_i}}, \quad \tilde{q}^*_{U_i} = \max_{m \in \{1,...,M\}} q^*_{U_{m_i}}. \tag{21}$$

The union ensemble method implies that at least one piece of information is considered to be true. The union operation in Eq. (21) has a significant limitation, as it results in an expanded credal set and introduces an overestimation effect on the precise union of credal sets, as shown in Figure 14.

**Intersection**  A collection of convex probability intervals $\{[\boldsymbol{q}^*_{L_m}, \boldsymbol{q}^*_{U_m}]\}^M_{m=1}$ can formulate an intersection as

$$\tilde{q}^*_{L_i} = \max_{m \in \{1,...,M\}} q^*_{L_{m_i}}, \quad \tilde{q}^*_{U_i} = \min_{m \in \{1,...,M\}} q^*_{U_{m_i}}. \tag{22}$$

However, the obtained $[\tilde{\boldsymbol{q}}^*_L, \tilde{\boldsymbol{q}}^*_U]$ does not inherently satisfy the condition outlined in Eq. (2) for constructing a credal set [17]. Therefore, the intersection approach is not applicable in CreDEs.

**Empirical evaluation**  In this experiment, we mainly evaluate the impact of averaging and union ensemble approaches on the EU estimation (GH($\mathbb{Q}$)) of CreDEs. Utilizing 15 individually trained ResNet50-based CreNets on CIFAR10 dataset, we formulate 15 CreDEs-M by varying the ensemble number M from 2 to 10 through averaging and union ensemble methodologies. Each kind of CreDEs-M is assessed for the averaged GH($\mathbb{Q}$) concerning samples and the quantity of CreDEs-M, and the averaged standard deviation (STD) of GH($\mathbb{Q}$) related to samples and the quantity of CreDEs-M. The results are plotted in Figure 15 (b) and (a), respectively. Besides, we also present the AUPRC and AUROC scores of OOD detection using GH($\mathbb{Q}$) as the uncertainty metric in Figure 15 (c) and (d), accordingly.

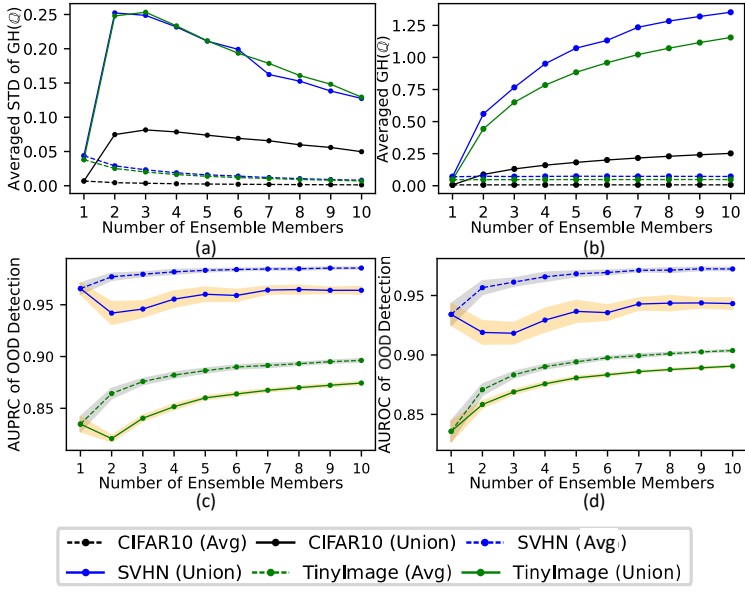

Figure 15: Impact of averaging (Avg) and union on the EU estimation of CreDEs on OOD detection benchmark involving CIFAR10 vs. SVHN/Tiny-ImageNet (TinyImage), implemented on ResNet50 architecture.

Figure 15 (a) illustrates a reduction in the averaged standard deviation (STD) of GH($\mathbb{Q}$) estimates as the number of ensemble members increases. This suggests that averaging the ensemble helps mitigate the uncertainty resulting from the randomness in the CreNet training process. Consequently, the AUROC and AUPRC scores, shown in Figure 15 (c) and (d), exhibit enhancement through the utilization of the averaging ensemble approach, accompanied by a concurrent reduction in the STDs of the scores as the number of ensemble members increases. In contrast, Figure 15 (b) highlights the overestimation of EU across various datasets when employing the union ensemble method. While the average EU estimates for ID samples are overall lower than those for OOD instances, the overestimation may lead to OOD mis-detection in some specific samples. This explains the "fluctuations" in the associated AUPRC and AUROC score curves in Figure 15.

# E  Further Discussion on Future work

## E.1  Generalizing Cross-Entropy for Lower/Upper Probability

As stated in Sec. 2.2.2, calculating the cross-entropy (CE) loss for the lower and upper probability works due to the 'one-hot' labeling nature of the ground truth vector $t$. However, generalizing the CE, which corresponds to the Kulback Leibler (KL) divergence

$$D_{KL}(\boldsymbol{t}|\boldsymbol{q}) = \sum_{j=1,...,C} t(j) \log \left( \frac{t(j)}{q(j)} \right)$$

between a predicted probability vector $\boldsymbol{q}$ and the general ground truth vector $\boldsymbol{t}$, to lower/upper probabilities is still an open research subject [68, 69].

In our case, the credal set $\mathbb{Q}$ is defined by the outputted proper probability intervals $[\boldsymbol{q}_L, \boldsymbol{q}_U]$. Therefore, the KL divergence for a lower probability inducing a credal set may be calculated by:

- Finding the probability vector that best approximates it. For probability intervals, there are two established such ways: normalizing either the lower or the upper probability (see [74]) or computing the so-called 'intersection probability' (see [15]).

- Computing the KL divergence between the ground truth vector and the approximation obtained.

In future work, we aim to investigate the approach and compare those other well-founded methods for calculating the cross-entropy loss with the one used in the paper.

## E.2  Theoretical Coverage Guarantees

In the current stage, our CreNets do not provide coverage guarantees, e.g., on how likely it is for the divergence of future data distributions to be within the modeled bounds. Nevertheless, various approaches to incorporating statistical guarantees in our framework can be envisaged.

In particular, a CreDE, being a classifier, can be employed as the 'underlying model' in an inductive *conformal learning* framework [64], which builds an empirical cumulative distribution of the 'non-conformity' scores of a set of calibration samples and at test time outputs the set of labels whose empirical CDF is above a desired significance level $\epsilon$ (e.g., 90%).

Namely, given a test input $x$ and the associated predictive system of probability intervals $[q_{L_c}, q_{U_c}], c = 1, ..., C$ (the output of CreDE), a sensible choice, for instance, is to set as non-conformity score of a pair (test input, class), $(x, c)$, the complement of the upper probability for that class, given input $x$:

$$s(x, c) \doteq 1 - q_{U_c}$$

(i.e., a label $c$ would be considered 'non-conformal' if its predicted upper probability, for that input $x$, is low), and compute predictive regions as standard in conformal learning:

$$\Gamma(x) = c \in \mathcal{C} : p^c > \epsilon,$$

where

$$p^c = \frac{|(x_j, c_j) : s(x_j, c_j) > s(x, c)|}{q + 1} + u \cdot \frac{|(x_j, c_j) : s(x_j, c_j) = s(x, c)|}{q + 1},$$

$(x_j, c_j)$ is the $j$-th calibration point, $q$ is the number of calibration points, and $u \sim \mathcal{U}(0, 1)$ (the uniform distribution on the interval $(0, 1)$).

We plan to explore this integration as the next step of our future work.

## E.3  Extension for Regression Framework

The vast majority of papers using credal sets in machine learning focus on classification [12, 5, 78], or, more recently, on self-supervised learning (but still in a classification setting [37]). Nevertheless, a recent study [19] has shown that the formalism of belief functions (a special class of credal sets) can

be extended to regression, by leveraging random fuzzy sets. It might thus be possible to explore such connections between probability intervals and random fuzzy sets and devise a suitable regression framework based on CreDEs.

The following section outlines a more direct possible extension of CreDEs to regression problems as a future research direction.

Remember that a CreNet outputs a credal set on the simplex of probability distributions over the classes. Each vertex of this credal set is therefore a probability distribution over the target space (the set of classes $C$ for classification).

On the other hand, a Bayesian regressor network (trained to learn a distribution of its weights) would output a (continuous) probability density over the target space (for the sake of simplicity, assume $Y = \mathbb{R}$).

One could then train an ensemble of Bayesian regressor networks to predict a credal set with a fixed number of vertices (one network outputting one vertex probability) so that the final predicted credal set is the convex closure of those. Figure 16 illustrates the concept briefly.

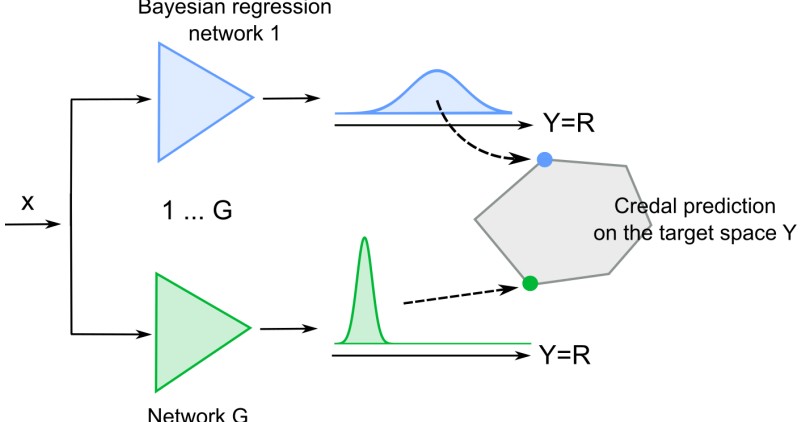

Figure 16: Concept of a credal regressor.

The Distributionally Robust Optimization (DRO) framework employed for CreDE training is to model the divergence between the data distribution of samples belonging to $G$ different groups within the training set. A full DRO formulation with $G$ loss components, in combination with Bayesian deep learning techniques such as variational inference, could then be employed to drive the training of the $G$ credal vertex networks, encouraged to generate diverse (probabilistic) predictions to model different possible data distributions, in a generalization of the two-component loss used here.

## Broader Impacts

The main objective of this paper is to advance the field of Machine Learning by improving the quality of uncertainty quantification. There are many potential societal consequences of our work, none of which we feel must be specifically highlighted here. The proposed method demonstrates superior performance in detecting out-of-distribution (OOD) samples. Such capability can potentially safeguard end users from misguided decisions that stem from the incorrect predictions of neural networks on OOD instances. Therefore, our approach can potentially improve the safety, reliability, and trustworthiness of machine learning systems for classification tasks and be applied in mission-critical domains, such as autonomous driving and medical sciences.

