# OpenReview forum: "Credal Deep Ensembles for Uncertainty Quantification"
_NeurIPS.cc/2024/Conference — NeurIPS 2024 poster_

### Official Review · Reviewer_YsPp · 2024-06-25

**Soundness:** 3
**Presentation:** 3
**Contribution:** 2
**Rating:** 5
**Confidence:** 4

**Summary:**

This paper presents Credal Deep Ensembles (CreDEs), an ensemble framework of Credal-Set Neural Networks (CreNets) to produce a high-quality epistemic uncertainty by using a credal set with the Distributionally Robust Optimization (DRO) technique.
Contributions are:
- A rigorous CreNet final layer in Section 2.1, where the number of nodes is doubled to present the confidence intervals with lower and upper probability bounds for each class. The final set of probability intervals is shown to be proper by satisfying convexity conditions.
- A training procedure with CreNet loss in Section 2.2, where the standard cross-entropy loss is trained with the upper-class probability bounds, and adversarially reweighted DRO is trained with the lower ones.
- Experimental results show CreDEs are better than Deep Ensembles (DEs) by having a higher test accuracy, lower ECE in IID classification, AUROC/AUPRC in OOD detection tasks.

**Strengths:**

- The paper is well-written and clear to understand. Sections are well organized, and the proposed method and its components are clear.
- I like the proposed CreNet final layer. I also like the motivation of this paper, the topic of improving uncertainty and robustness performance in Neural Networks (NNs) is important and has broad applications in the AI safety field.
- The empirical evaluation covers different task performances in multiple experimental settings.

**Weaknesses:**

1. The theoretical contributions of this paper are weak. There are no theories to show the proposed method and the loss function can improve the uncertainty or robustness quality. The only theoretically sound result is showing the results in Eq. 4 satisfy convexity conditions in Eq. 2, but this can be derived straightforwardly by using the sum of standard exponential functions for the output vector.
2. Regarding the practical novelty, I also raise a concern about the computational limitation of the proposed method. Because DEs has been raised as computationally inefficient in estimating the NN uncertainty field [1, 3, 4, 5], CreDEs is even less efficient than DEs. In particular, for each ensemble member, in training time, the complexity of Alg. 1 with sorting and the training objective function is more complex than Standard Neural Network (SNN). In test time, the model also requires doubling the number of output nodes than SNNs and needs to solve two optimization problems in Eq. 13, causing a higher latency.
3. The proposed method only works for classification tasks. Additionally, it is also a heuristic and depends on hyperparameter tuning. In particular, choosing the adversarial weights $\displaystyle w$ for DRO is non-trivial and can significantly change the training objective function. Furthermore, choosing the number of class $K$ in Alg. 2 is also heuristic, causing a trade-off between uncertainty estimation quality and computational efficiency.
4. Lack of empirical comparison with many related works in uncertainty baselines [1]. I would be happy to raise my initial score if important baselines will be added, including MC Dropout [2], BatchEnsemble [3], Rank-1 BNN [4], MIMO [5], SNGP [6], and NatPN [7].
5. Lack of uncertainty evaluations and comparisons (see Questions). Better performance in OOD detection or active learning tasks is only a necessary condition, it is not a sufficient condition for a high-quality epistemic uncertainty algorithm.
6. The training complexity of CreNet is considerably higher than SNN in L-712, causing CreDEs's complexity to be even much higher than DEs by linearly increasing the number of ensemble sizes. The model inference complexity in Tab. 6 is not evaluated in different model architectures across different hardware settings, e.g., GPU, CPU, etc.

**Questions:**

1. Can you qualitatively evaluate the uncertainty quality by using the reliability diagrams [8] and the PDF plot of predictive entropy $\mathrm{H}(p(y|x))$ like Fig. 3 in [9]?
2. Regarding the quantitative evaluation of the epistemic uncertainty in Table 5: How about the performance of DEs? Can you add an evaluation by calculating $Var_i[\mu_i(x)]$ in [10]?
3. Regarding the robustness performance, can you compare your method with DEs regarding the Negative log-likelihood, test accuracy, and ECE on OOD data (e.g., train on CIFAR-10, test on CIFAR-10-C)?
4. Regarding Table 1: What is the number of $K$ used for this setting? Could you please specify why there are significant differences in your reported ECE with [1] on ImageNet about DEs?
5. I am also curious about the results of different settings of DEs. For instance, the DEs used in your paper is the mixture of experts (or operation), how about the product of experts (and operation)?

References:

[1] Nado et al., Uncertainty Baselines: Benchmarks for Uncertainty & Robustness in Deep Learning, arXiv preprint arXiv:2106.04015, 2021.

[2] Gal et al., Dropout as a Bayesian Approximation: Representing Model Uncertainty in Deep Learning, ICML, 2016.

[3] Wen et al., J. Batchensemble: an alternative approach to efficient ensemble and lifelong learning, ICLR, 2020.

[4] Dusenberry et al., Efficient and scalable Bayesian neural nets with rank-1 factors, ICML, 2020.

[5] Havasi et al., Training independent subnetworks for robust prediction, ICLR, 2021.

[6] Liu et al., Simple and principled uncertainty estimation with deterministic deep learning via distance awareness, NeurIPS, 2020.

[7] Charpentier et al., Natural posterior network: Deep bayesian predictive uncertainty for exponential family distributions, ICLR, 2022.

[8] Guo et al., On Calibration of Modern Neural Networks, ICML, 2017.

[9] Lakshminarayanan et al., Simple and scalable predictive uncertainty estimation using deep ensembles, NeurIPS, 2017.

[10] Valdenegro-Toro et al., A Deeper Look into Aleatoric and Epistemic Uncertainty Disentanglement, CVPR Workshop, 2022.

---

**After rebuttal.** The authors provide detailed additional results, and I believe that they are valuable to support the claim of the main paper. So, I increased my initial rating for this paper.

**Limitations:**

Please see my comments on the Weaknesses.

---

> ### Author Rebuttal · Authors · 2024-08-07
>
> Dear Reviewer,
>
> We thank you for your valuable feedback.
>
> **Response to W1:** We believe that lines 172-175 in the original paper are also an important result, which shows that cross-entropy loss (CE) can be applied to upper probability vectors as representatives of part of the boundary of the predicted credal set.
>
> However, in the original paper, indeed, the only validation of our hypothesis that credal prediction leads to better accuracy, calibration, and OOD detection is empirical. Below we outline a more rigorous argument.
>
> Epistemic uncertainty in predictions is induced by the probability distribution(s) that generated the training data may be very different from the distribution that generates a test point. It is well known that using CE as a loss leads to the maximum likelihood estimator of the model parameters, under the assumption that all training points are drawn i.i.d. from a single, unknown distribution $P$:
> $$
> \hat{\theta} = \arg \min_\theta\text{CE} = \theta_{MLE}.
> $$
>
> What does Distributionally Robust Optimization (DRO) do? It takes a cautious stance, and assumes training points are generated by one of many potential distributions $\mathcal{U}$, and minimizes the upper bound of the expected risk (Eq. (6) of the paper). In practice, this is approximated using Adversarially Reweighted Learning as in Eq. (9). Therefore, it learns
> 			$\hat{\theta} = \arg \min_{P \in \mathcal{U}} E_P (Loss)$.
> However, if we apply DRO to a traditional network, the network will learn a single parameter value (Eq. (9) of the paper); for a new test data point $x_t$ it will then produce a single predicted probability vector, the one produced by the model that, in the long run, minimizes the worst-case risk.
>
> This, though, does not express at all the (epistemic) uncertainty about *which* of the distributions in $\mathcal{U}$ has generated the particular test datapoint $x_t$; as a result, it does not express how uncertain the prediction may be for that particular data point.
>
> Allowing a network to output lower and upper probability vectors for each input, instead, allows this epistemic uncertainty to be explicitly modeled.
>
> What is the best-case scenario? That in which all data (including the test point $x_t$) are indeed generated by the same distribution – in this case, $\theta_{MLE}$ is the best choice, in a maximum likelihood sense.
>
> What is the worst case? That in which $x_t$ is generated by a distribution that is far away from the groups of distributions $P_g$ (see lines 140-141) that generated the training data. In this case, the DRO solution will still be far from perfect, but still the best one can do, given the evidence in the training data.
>
> Therefore, if I minimize the CE versus $\boldsymbol{q}_U$ (first component of Eq. (10)), I get a credal set Cre of predictions whose upper probability is such that, for each class $i = 1,…,C$, there exists a probability vector $\boldsymbol{q}\in \mathbb{Q}$ such that $q(i)$ is the MLE prediction for that class (please remember the discussion of lines 172-175).
>
> Similarly, if I minimize the DRO loss versus $\boldsymbol{q}_L$ (the second component of Eq. (10)), for each class $i = 1,…,C$, there exists a probability vector $\boldsymbol{q} \in \mathbb{Q}$ such that $q(i)$ is the DRO prediction for that class.
>
> Minimizing the two components together, as in Eq. (10), thus encourages the predicted credal set to extend from the best-case to the worst-case prediction.
>
> **Response to W2, W4, and W6**: We kindly invite you to follow the detailed response in our global rebuttal. Based on your suggestions, we further reported additional inference complexity comparison in Table 1 of the rebuttal file and included MC Dropout and two TensorFlow-standardized BNNs.
>
> **Response to W3**: Yes, our method is for classification and we provide a roadmap for extending it to regression tasks in Appendix E.3. Our training includes a hyperparameter $\delta$. We performed the ablation study (lines 287-312), showing that our CreDEs are not particularly sensitive to $\delta$. $K$ is a design choice by the end user. The ablation study in Appendix B.2 shows that increasing the value of $K$ improves performance and increases cost. In addition, the reduction algorithm has a strong mathematical background in probability intervals [17], which can rigorously guarantee a valid credal set.
>
> **Response to W5, Q1, Q2**: Based on your advice, we reported the entropy plots and reliability diagrams in the rebuttal file. Regarding Table 5, the EU estimates of DEs-5 are 0.0996 (CIFAR10), 0.4758 (SVHN), and 0.4753 (Tiny-ImageNet). Although the EU estimates of DEs-5 and CreDEs-5 are not comparable in values due to different representations, CreDEs-5 exhibits much higher EU estimates of OOD samples than those of ID data and more reliable predictions on ID samples, qualitatively. We could not report $Var_i[\mu_i(x)]$ in our cases because the calculation is for models with a custom softmax layer, according to Eq. (6-8) in [10].
>
> **Response to Q3**: In our settings, the corrupted CIFAR10 samples will be detected as OOD data and be to abstain. Therefore, we did not report the accuracy and ECE values on CIFAR10-C. For completeness, we reported the numbers in Table 3 of the rebuttal file, showing the visibly improved performance of CreDEs compared to DEs.
>
> **Response to Q4**: $K$ is used to reduce the complexity of uncertainty estimation. We do not employ Alg. 2 for accuracy and ECE calculation. The difference is most likely because the training settings are significantly different. As the ImageNet experiments are conducted fairly for DEs and CreDEs (shown in Appendix C), we believe the results can properly reflect the performance improvements.
>
> **Response to Q5**: We randomly selected five members from the 15 models to construct the ensemble. The member list of each ensemble was strictly different. Could you please specify the settings that you are curious about? We are eager to continue the discussion.

---

> > ### Comment · Reviewer_YsPp · 2024-08-07
> > **Clarification for Question 5**
> >
> > Thanks for your response. Regarding Q5, I am curious about the ensemble setting with the "product of experts" [1], not "mixture of experts" setting that is currently used in your paper.
> >
> > References:
> >
> > [1] Hinton, G.E, Products of experts, 9th International Conference on Artificial Neural Networks, 1999.

---

> > > ### Author Response · Authors · 2024-08-08
> > > **Further discussion on Question 5**
> > >
> > > Dear Reviewer,
> > >
> > > Thank you for your engagement in the discussion.
> > >
> > > Following your enlightenment, we conducted the ablation study over deep ensembles using the product of expert strategy, denoted as DEs-5 (POE).
> > >
> > > Table 1 reports the test accuracy (ACC) and ECE values on the CIFAR10 dataset of different models across ResNet50, VGG16, and ViT Base architectures.
> > >
> > > Table 1. Test accuracy (ACC) (%)and ECE comparison.
> > > |                            |  ResNet50: ACC |   ResNet50: ECE   |   VGG16: ACC   |     VGG16: ECE    |  ViT Base: ACC |   ViT Base: ECE   |
> > > |:--------------------------:|:--------------:|:-----------------:|:--------------:|:-----------------:|:--------------:|:-----------------:|
> > > | CreDEs-5 ($\hat{i}_{max}$) | **93.74±0.11** | **0.0109±0.0017** | **87.92±0.11** | **0.0611±0.0012** | **93.59±0.39** | **0.0104±0.0012** |
> > > | CreDEs-5 ($\hat{i}_{min}$) | **93.75±0.11** | **0.0092±0.0016** | **87.94±0.11** | **0.0203±0.0014** | **93.60±0.40** | **0.0107±0.0014** |
> > > |            DEs-5           |   93.32±0.13   |   0.0131±0.0010   |   85.53±0.10   |   0.0815±0.0011   |   90.43±0.97   |   0.0181±0.0019   |
> > > |         DEs-5 (POE)        |   93.47±0.11   |   0.0610±0.0011   |   85.55±0.08   |   0.1368±0.0008   |   90.56±0.90   |   0.0894±0.0087   |
> > >
> > > *Table 1 shows that DEs-5 (POE) could improve test accuracy but significantly reduce the calibration performance of DEs-5 (larger ECE values). In these comparisons, our CreDEs-5 is the most superior method.*
> > >
> > > In addition, we also present the Reliability diagram in the table format in Table 2. Table 2 can equally reflect the results of CreDEs-5 and DEs-5 in Figure 3 of the rebuttal file.
> > >
> > > Table 2. Test accuracy (ACC) of the ResNet50-based models in each confidence bin of the reliability diagram.
> > >
> > > |      Confidence Bin      | [0.0, 0.1] | [0.1, 0.2] | [0.2, 0.3] | [0.3, 0.4] | [0.4, 0.5] | [0.5, 0.6] | [0.6, 0.7] | [0.7, 0.8] | [0.8, 0.9] | [0.9, 1.0] |
> > > |:--------------------------:|:----------:|:----------:|:----------:|:----------:|:----------:|:----------:|:----------:|:----------:|:----------:|:----------:|
> > > | CreDEs-5 ($\hat{i}_{max}$) |     0.0    |     0.0    |   0.3333   |   0.1951   |   0.4206   |   0.5492   |   0.6361   |   0.7532   |   0.8849   |   0.9863   |
> > > | CreDEs-5 ($\hat{i}_{min}$) |     0.0    |     0.0    |   0.3333   |   0.2759   |   0.4688   |   0.5701   |   0.6444   |   0.7673   |   0.8899   |   0.9872   |
> > > |            DEs-5           |     0.0    |     0.0    |     0.5    |   0.2857   |   0.4031   |   0.5035   |   0.6569   |   0.7092   |   0.8583   |   0.9839   |
> > > |         DEs-5 (POE)        |     0.0    |     0.0    |     0.0    |     0.0    |     0.0    |     0.3    |   0.3947   |     0.5    |   0.3529   |   0.9396   |
> > >
> > > *Table 2 demonstrates the worse performance of DEs-5 (POE) from the reliability perspective. Our model performs best out of the three.*
> > >
> > > Furthermore, we also study the uncertainty estimation performance via OOD detection. Here, we computed the entropy of the final prediction of DEs-5 (POE) as the total uncertainty. In the cases of CreDEs-5 and DEs-5, we use the upper entropy $\overline{H}(\mathbb{Q})$ and $H(\tilde{\boldsymbol{q}})$ as in our paper, respectively. *The results in Tables 3, 4, and 5, consistently show the outperformance of our method.*
> > >
> > > Table 3: OOD detection performance comparison (%) on ResNet50 architecture.
> > >
> > > |             | CIFAR10 vs. SVHN | CIFAR10 vs. SVHN | CIFAR10 vs.Tiny-ImageNet | CIFAR10 vs.Tiny-ImageNet |
> > > |:-----------:|:----------------:|:----------------:|:------------------------:|:------------------------:|
> > > |             |      AUROC       |      AUPRC       |          AUROC           |           AUPRC          |
> > > |   CreDEs-5  |    **95.71±0.42**    |    **97.73±0.27**    |        **89.02±0.10**        |        **88.02±0.15**        |
> > > |    DEs-5    |    94.80±0.43    |    97.26±0.29    |        88.80±0.19        |        87.21±0.29        |
> > > | DEs-5 (POE) |    93.90±0.24    |    96.10±0.21    |        88.03±0.20        |        84.11±0.32        |
> > >
> > > Table 4: OOD detection performance comparison (%) on VGG16 architecture.
> > >
> > > |             | CIFAR10 vs. SVHN | CIFAR10 vs. SVHN | CIFAR10 vs.Tiny-ImageNet | CIFAR10 vs.Tiny-ImageNet |
> > > |:-----------:|:----------------:|:----------------:|:------------------------:|:------------------------:|
> > > |             |      AUROC       |      AUPRC       |          AUROC           |           AUPRC          |
> > > |   CreDEs-5  |    **87.05±0.80**    |    **93.36±0.42**    |        **82.14±0.14**        |        **80.81±0.16**        |
> > > |    DEs-5    |    84.50±0.49    |    90.78±0.35    |        79.40±0.10        |        75.91±0.14        |
> > > | DEs-5 (POE) |    84.10±0.22    |    89.83±0.16    |        78.11±0.08        |        72.23±0.16        |

---

> > > > ### Author Response · Authors · 2024-08-08
> > > > **Further discussion on Question 5**
> > > >
> > > > Table 5: OOD detection performance comparison (%) on VGG16 architecture ViT Base.
> > > >
> > > > |             | CIFAR10 vs. SVHN | CIFAR10 vs. SVHN | CIFAR10 vs.Tiny-ImageNet | CIFAR10 vs.Tiny-ImageNet |
> > > > |:-----------:|:----------------:|:----------------:|:------------------------:|:------------------------:|
> > > > |             |      AUROC       |      AUPRC       |          AUROC           |           AUPRC          |
> > > > |   CreDEs-5  |  **87.30±1.77**  |  **92.24±1.15**  |      **88.17±0.44**      |      **86.94±0.60**      |
> > > > |    DEs-5    |    79.80±1.75    |    87.97±1.17    |        83.81±0.81        |        81.67±0.89        |
> > > > | DEs-5 (POE) |    82.41±1.56    |    88.51±0.95    |        83.21±1.02        |        78.24±1.17        |
> > > >
> > > > *The ablation study shows again the outperformance of our proposed CreDEs-5. The product of expert strategy improves the accuracy of classical DEs, but significantly damages the calibration performance and overall leads to worse OOD detection performance.*
> > > >
> > > > We deeply thank you again and hope the increased clarity in our responses has addressed your questions.

---

> > > > > ### Comment · Reviewer_YsPp · 2024-08-10
> > > > >
> > > > > I thank the authors for the detailed response. I appreciate additional results from the authors, and as promised, I have raised my initial rating. However, there are still concerns, including:
> > > > >
> > > > > **1. From the theory point of view, the contribution is still weak.**
> > > > >
> > > > > The intuition answer in the rebuttal for W1 doesn’t’ address my concern. If your model outperforms DEs in practice, e.g., in terms of robustness, I expect the authors to show a generalization bound or robustness certificate. This will show whether your model is generally better than DEs, or only in some specific cases.
> > > > >
> > > > > **2. CreDEs is even more computationally inefficient than other baselines.**
> > > > >
> > > > > Regarding the computational limitation in W2-4-6,
> > > > >  - Firstly, in Tab.1-PDF, as mentioned in W6, I think more hardware settings should be considered when evaluating computational efficiency. For instance, in powerful resources, CreDEs can be slightly slower than SNNs. But for smaller hardware (lower GPU memory, smaller CPUs), CreDEs can be significantly slower. This evaluation will reflect where CreDEs can be deployed in the real world (e.g., servers, mobiles, IoT devices, etc.).
> > > > > - Secondly, comparisons with BNN-R, and BNN-F are not convincing to me. They are quite outdated methods. For instance, in the literature on BNNs, please check Rank-1 BNN (others in my citations in W4). I understand that all of them might not be as good as DEs, but they are more computationally efficient than DEs and CreDEs. Adding them to the comparison, I believe will help the community know which model they should use in specific cases.
> > > > >
> > > > > **3. CreDEs is heuristic and depend on hyperparameter tuning.**
> > > > >
> > > > > Regarding the heuristic limitations in W3, I am still concerned about the additional hyperparameter selection. I think that they are not trivial in practice. Specifically,
> > > > > - In Tab.5, I still see significant differences between $\delta=0.5$ and $\delta=96875$. Also, this table only shows EU estimates, but I believe it also changes the accuracy by $\delta$ related to the DRO objective function.
> > > > > - Regarding the trade-off limitation of the selection of $K$, the rebuttal “increasing the value of improves performance and increases cost” is vague. Increasing the cost is clearly not good in practice.

---

> ### Author Response · Authors · 2024-08-13
> **Further responses for remaining questions**
>
> Dear Reviewer,
>
> We express our sincere appreciation for your recognition of our additional results and your effort in reviewing our paper. Please allow us to address your remaining concerns.
>
> **Lacking of a generalization bound or robustness certificate from a theoretical perspective**
>
> Thank you for clarifying your question. This is true, CreDEs currently do not provide coverage guarantees/robustness certificates. However, the extensive experiments across different datasets, architectures, and tasks do empirically show that our CreDEs generally achieve consistent improvements over DEs. We have acknowledged lacking theoretical guarantees of our CreDEs in our conclusion (lines 349-350) and detailed our future research plan in Appendix E.2. Incorporating your comments, we will expand our emphasis on this point in the main body of the revised paper.
>
> **Computationally inefficient than other baselines**
>
> *Regarding more hardware settings to evaluate the inference cost*
>
> Following your comments, we report the inference cost measured on a single AMD EPYC 7643 48-core CPU in Table R1. From the result in Table R1, we did not observe any significant overhead. In addition, we can also observe that using VGG16 (a lighter model compared to ResNet50) can visibly reduce the inference cost of DEs and CreDEs.
>
> Table R1: Inference cost comparison between SNN and CreNet per single CIFAR10 input of different architectures.
>
> |        | VGG16 (ms) | ResNet50 (ms) |
> |:------:|:----------:|:-------------:|
> |   SNN  | 19.2 ± 3.8 |  148.2 ± 49.0 |
> | CreNet | 23.1 ± 5.2 |  163.3 ± 39.4 |
>
> Since our main evaluation scope focuses on image classification of different deep neural network architectures across various tasks, good-performance GPUs are a favorable choice for implementation.
>
> We appreciate your consideration of the complexity of applying our models on smaller hardware such as mobile and IoT devices. To our knowledge, in the context of smaller hardware, lightweight neural networks are generally used, and there are also techniques such as quantization to further improve computational efficiency.  We believe that these factors, as well as optimized code implementation of our CreDEs, could help mitigate the added complexity compared to DEs. *If practitioners are already using or planning to use DEs in their applications, we believe it is worth considering our CreDEs to improve performance with an acceptable increase in complexity.*
>
> *Regarding further comparisons.*
>
> We fully agree that there are uncertainty-aware models that are more computationally efficient than DEs and CreDEs. As you rightly pointed out, achieving comparable or slightly lower performance of DEs while being computationally cheaper is the main objective of this line of research. However, since DEs are active strong baselines [1, 2] and are actively applied in practice, such as [3] [4] [5], we aim to improve the performance of DEs (e.g., the quality of uncertainty estimation) to benefit safety-critical applications in particular.
>
> Following your comments, we tried to implement the Rank-1 BNNs in our environment to fairly compare the performance between Rank-1 BNNs and CreDEs. Implementing Rank-1 BNNs requires installing a package of Edward2's Bayesian Layers from their official GitHub repository. Unfortunately, although we followed the instructions, we could not import the edward2 package correctly due to dependency conflicts, software version incompatibility, and limited time. There are the same unresolved issues posted by others on the official GitHub repository.
>
> Nevertheless, from the main results of Tables 1-4 in the original paper on Rank-1 BNNs, we could observe that Rank-1 BNNs achieve comparable or slightly lower performance compared to DE baselines while being computationally efficient. Combined with our observations in our paper that CreDEs generally outperform DEs, we believe, there is a high probability that our CreDEs could achieve better performance than Rank-1 BNNs.
>
> The practical message we could deliver to the community is that our CreDEs have a high potential to improve the performance of DEs in real-world applications with an acceptable increase in complexity. If DEs are already out of consideration in practice due to computational constraints, our CreDEs would indeed not be an ideal option.
> As promised, we will extend our original discussion on the computational limitations of CreDEs (lines 347-348) in the revised paper.
>
> [1] Benchmarking uncertainty disentanglement: Specialized uncertainties for specialized tasks. 2024.
>
> [2] Deep deterministic uncertainty: A new simple baseline. CVPR 2023.
>
> [3] De-tgn: Uncertainty-aware human motion forecasting using deep ensembles. IEEE Robotics and Automation Letters 2024
>
> [4] Flood uncertainty estimation using deep ensembles. Water, 2022.
>
> [5] Benchmarking common uncertainty estimation methods with histopathological images under domain shift and label noise. Medical image analysis, 2023.

---

> ### Author Response · Authors · 2024-08-13
> **Further responses for remaining questions**
>
> **hyperparameter $\delta$**
>
> Throughout the main experimental evaluations (except the ablation study on $\delta$), we set the default $\delta=5$, to reflect a balanced assessment of the train-test divergence and show how such a value allows our model to outperform the baselines. Upon this setting of $\delta$, the extensive comparisons consistently show our CreDEs outperform DEs.
>
> The ablation study on Hyperparameter $\delta$ (lines 287-312) verifies the robustness of CreDEs across hyperparameter setups in terms of test accuracy, ECE, and OOD detection performance, consistently better than the performance of DEs. For a visible sense, we copied the results of Table 4 of the paper as follows in Table R2.
>
> Table R2: Performance comparison of CreDEs trained on the CIFAR10 dataset under different settings of $\delta$.
>
> |                     |                 |   0.5  |  0.625 |  0.75  |  0.875 | 0.9375 | 0.96875 |
> |:-------------------:|:---------------:|:------:|:------:|:------:|:------:|:------:|:-------:|
> |     Test ACC (%)    | $\hat{i}_{max}$ |  93.74 |  94.54 |  94.47 |  94.57 |  93.88 |  93.99  |
> |                     | $\hat{i}_{min}$ |  93.75 |  94.55 |  94.47 |  94.56 |  93.87 |  93.99  |
> |      ECE Values     | $\hat{i}_{max}$ | 0.0092 | 0.0059 | 0.0053 | 0.0056 | 0.0087 |  0.0086 |
> |                     | $\hat{i}_{min}$ | 0.0109 | 0.0087 | 0.0089 | 0.0093 | 0.0087 |  0.0087 |
> |      SVHN (OOD)     |      AUROC      |  97.44 |  97.44 |  97.92 |  97.95 |  97.42 |  97.51  |
> |                     |      AUPRC      |  93.07 |  96.34 |  97.00 |  96.92 |  98.79 |  98.82  |
> | Tiny-ImageNet (OOD) |      AUROC      |  88.28 |  89.01 |  89.10 |  89.18 |  89.95 |  89.24  |
> |                     |      AUPRC      |  88.13 |  89.81 |  89.76 |  89.72 |  89.18 |  89.26  |
>
>
> Table 5 of the paper reports the averaged EU estimate under different settings of $\delta$. You are correct, we could see significant differences between $\delta=0.5$ and $\delta=96875$ (The EU estimates using $\delta=96875$ are much smaller than those using $\delta=0.5$).
>
> From Table 5, we could also observe that increasing the value of $\delta$ (i.e., giving less importance to the divergence between test and training distributions) leads to a decreasing trend in the average EU estimates per dataset (particularly for ID CIFAR10 samples). This aligns with the intuition that, if the model is more uncertain about the divergence of the distributions (smaller $\delta$), it should express a larger EU. Despite smaller uncertainty values at high $\delta$’s, the difference between In-domain (ID) and OOD samples remains noticeable. This explains why a $\delta$ closer to 1 does not always lead to low-performance OOD detection and why our model’s OOD detection performance is robust against the choice of $\delta$.
>
> The ablation study also shows that we further improve the performance of CreDEs by finding the 'best' $\delta$. One possible way is to conduct standard cross-validation on specific test scenarios. Considering the performance of CreDEs robust against $\delta$, an alternative interesting option, in the presence of multiple datasets (e.g., acquired over time in a continual learning setting), could be applying the DRO loss component to different components of the training set, and assessing the results to robustly select $\delta$. (lines 294-300).
>
> **Regarding the trade-off parameter $K$**
>
> The parameter $K$ represents a trade-off between uncertainty estimation and computational efficiency. As demonstrated in the ablation study in Appendix B.2,  $K\leq10$ is a practical range for measuring epistemic uncertainty using the Generalized Hartley (GH) measure. As shown in Tables 7 and 8 and Figure 7 of Appendix B.2, the application of $K=4$ for OOD detection involving the CIFAR10 dataset (reducing the dimension from 10 to 4 classes and costing 0.02 ms per input sample) and the application of $K=10$ for the OOD detection involving the CIFAR100 datasets (reducing the dimension from 100 to 10 classes and costing 17 ms per input sample) can guarantee the outperformance of CreDEs over the DEs.
>
> Choosing $K$ for computing the upper and lower entropy measures is more flexible. For instance, in Tables 2 and 7 and Figure 8 of the paper, $K=20$ for the ImageNet experiment dataset (reducing the original 1k to 20 classes and costing 6.1ms per sample) shows an improved OOD detection performance compared to DEs.
>
> We appreciate your efforts to review our work and your active engagement in this discussion. We benefit from the process. We hope that the increased clarity of our responses has addressed your concerns.

---

> > ### Comment · Reviewer_YsPp · 2024-08-13
> >
> > Thank you for your explanation. I also thank the authors for clarifying the ablation study of hyperparams $\delta$ regarding test accuracy. Most of my concerns are addressed. Because of theoretical contribution and computational limitations,  I updated my final score to 5. Good luck!

---

> > > ### Author Response · Authors · 2024-08-14
> > >
> > > Dear reviewer,
> > >
> > > Thank you for your positive feedback. We appreciate the time and effort you have dedicated to reviewing our paper and the discussions!

---

### Official Review · Reviewer_7GAw · 2024-06-27

**Soundness:** 4
**Presentation:** 4
**Contribution:** 4
**Rating:** 7
**Confidence:** 4

**Summary:**

The paper proposes credal neural networks, which output probability intervals for each class as opposed to a single probability estimate. They also propose to use ensembles of these models, averaging the outputs of the members. Their ensembles of credal neural networks show a higher performance than traditional ensembles in various experiments, from epistemic uncertainty (OOD detection), to the normal accuracy on ImageNet.

**Strengths:**

The method is benchmarked across an impressive range of tasks and datasets in both main paper and appendix and shows consistent improvements over traditional deep ensembles.

The method has many hidden details, e.g., trivial solutions it could collapse to. The paper uncovers many of these potential problems and explains how they are solved by neat implementation tweaks. These tweaks can be relevant to other credal approaches.

Reproducible code is provided, along with pseudo code in the paper.

**Weaknesses:**

In all experiments, there is only one baseline, namely traditional deep ensembles. I would encourage to pull results for the standard ensemble with DRO loss (Called DEs*-5) from the appendix to the main paper and to also implement other methods, like Mahalanobis (https://arxiv.org/abs/1807.03888), which recently showed SOTA performance on EU estimation (https://arxiv.org/pdf/2402.19460)

The resampling setup is internally correlated as the authors construct their 15 ensembles by drawing 5 members out of 15 trained models repeatedly. The 15 ensembles thus overlap which will likely lead to underestimating the error bars in the results

**Questions:**

To calculate accuracy, did you use i_max or i_min?

Do you have some insights into how big the credal intervals are usually, etc? I’m asking because I want to rule out the alternate hypothesis that your method is producing almost-Dirac credal sets.

I understand your motivation of why you apply the vanilla cross entropy loss to the upper bound estimate of the credal set $q_U$ as opposed to the midpoint $m$, but do you have ablations quantitatively supporting this choice?

Do the computational complexity studies in Table 6 include the constrained optimization search for maximum / minimum entropy? In Appendix C it seems like this could be a burden, where training a single ensemble credal ensemble member takes five times as long as training a standard cross-entropy loss member.

There’s a small typo in line 284, “AUORC”

**Limitations:**

The authors acknowledge that the runtime and RAM costs for ensembles are a hindering aspect of their work.

---

> ### Author Rebuttal · Authors · 2024-08-07
>
> Dear Reviewer,
>
> We sincerely appreciate your high recognition of our work and valuable feedback and suggestions. Below we address your concerns.
>
> **Regarding W1**
>
> **Response:** We fully agree that moving the additional results of DEs$^*$-5 (standard ensemble with DRO loss) from the appendix to the main body is a better presentation. Following your suggestion, we include the MC Dropout model in Table 2 of the rebuttal file, which is claimed to be a good choice for uncertainty quantification across the board in the evaluation of [A]. The additional comparison further demonstrates the outperformance of our CreDEs.  We did not report the result in the rebuttal because the Mahalanobis method is a distance-based method for uncertainty quantification and was outside the original scope, which was to address uncertainty quantification using ''second-order'' representations. However, we will properly extend the above discussion in the limitations section of the revised paper. For a more detailed response to the baseline choice, we invite you to read our global rebuttal (part 2).
>
> **Regarding W2**
>
> **Response:** In our experiments, we randomly selected five members from the 15 trained models to construct the ensemble. The ensemble member lists of each ensemble were strictly guaranteed to be different. Therefore, we believe that this setting could reflect the error bars of the results. Based on your comments, we will highlight the setting to improve clarity.
>
> **Regarding Q1**
>
> **Response:** Yes, we reported the accuracy of the models using $i_{max}$ and $i_{min}$ in Table 1 and Table 3 of the paper. They both show improvement in test accuracy.
>
> **Regarding Q2**
>
> **Response:** Due to the high dimensionality, it is hard to visualize and compute the size of the credal set when there are $C>3$ elements. However, we can indirectly judge whether the generated credal set resembles a Dirac credal set by calculating the maximum reachable upper bound probability of each sample, namely $\max{(q^*_{U_1}, ..., q^*_{U_C})}$. The closer the probability value is to 1, the closer it is to an almost Dirac credal set. We show the results of ResNet50-based CreDEs-5 on the CIFAR10, SVHN, and Tiny-ImageNet datasets in *Figure 1 of the rebuttal file*. We can conclude that our method does not always generate almost Dirac credal sets, especially on OOD samples. For CIFAR10 samples, there are a large number of quasi-Dirac credal sets (but not all), which is reasonable considering the high test accuracy of CreDEs and the low ECE (shown in Table 1 of the paper).
>
> **Regarding Q3**
>
> **Response:** If we first calculate the probability vector based on the midpoint $\boldsymbol{m}$ and then calculate the cross-entropy loss, we observed that we could not make the model reasonably learn to generate half-length $\boldsymbol{h}$ to generate the probability interval from the training process, as the $\boldsymbol{h}$ is not incorporated in the training. In another case, we could not calculate the midpoint of the generated probability intervals as $\boldsymbol{q}_U-\boldsymbol{q}_L$ for cross-entropy calculation, it is not normalized. In Appendix E.1, we discuss another sensible way of generalizing cross-entropy for lower/upper probability as our future exploration.
>
> **Regarding Q4**
>
> **Response:** No, the inference cost in Table 6 does not include the uncertainty calculation cost. We reported the cost of calculating the generalized Hartley (GH) measure and the upper entropy in Figures 7 and 8 of the paper. For instance, the time cost for GH calculation for CIFAR10 without approximation is 17 ms (0.02 ms in the reduced case considering 4 out of 10 classes) while calculating EU in deep ensembles for CIFAR10 takes $4.1\times 10^{-4}$ ms, measured on the same single CPU. Though higher CreDEs remain practical without actual computational constraints. Besides, the numbers reported are without code efficiency optimization: a more efficient code implementation could significantly reduce the cost.
>
> Regarding the training complexity, the CreNet uses a custom training loop, unlike the TensorFlow-standardized training standard neural networks, precluding a fair comparison. Given the evidence that CreNets only marginally increases the inference time (single forward pass) in Table 6, we are optimistic that by standardizing and optimizing the custom training loop and adopting a more efficient code implementation of Algorithm 1, the training effort could be significantly reduced.
>
> **Regarding Q5**
>
> **Response:** Thank you for pointing the typo out. We will revise it accordingly.
>
> [A]  Benchmarking uncertainty disentanglement: Specialized uncertainties for specialized tasks. 2024

---

> > ### Comment · Reviewer_7GAw · 2024-08-12
> >
> > Thank you for your responses, especially Figure 1 of the rebuttal PDF. Given that this will be included in the revised paper and that the runtime analysis may get an improvement (like testing the traditional training in your custom training loop, for a fair comparison), I remain with my score of 7.

---

> > > ### Author Response · Authors · 2024-08-13
> > >
> > > Dear Reviewer,
> > >
> > > We are grateful for your efforts and valuable comments and suggestions in this review process. We promise to include the additional results (e.g. Figure 1 of the PDF) as suggested by you in the revised paper.
> > >
> > > Once again, we thank you for recognizing the quality of our work and supporting the acceptance of our paper.

---

### Official Review · Reviewer_AC8P · 2024-07-12

**Soundness:** 2
**Presentation:** 3
**Contribution:** 2
**Rating:** 5
**Confidence:** 3

**Summary:**

This paper introduces Credal Deep Ensembles, which are ensembles of Credal-Set Neural Networks designed to predict lower and upper probability bounds for each class, representing epistemic uncertainty.

**Strengths:**

Novel approach to uncertainty quantification in deep learning. Empirically it seems that the proposed approach is performing not too bad, while I am not entirely sure about the interpretation of the results (a lot of the results seem to indicate only marginal improvement).

**Weaknesses:**

Empirical results are not very convincing, since the improvements (if there are any) seem to be of marginal nature.

While the paper introduces novel methods and validates them empirically, it could benefit from a deeper theoretical analysis of why the proposed methods improve uncertainty quantification. E.g. the paper introduces novel components like the Interval SoftMax function and the application of DRO, but it lacks a detailed theoretical analysis that justifies their effectiveness.

Lack of baseline comparisons (some model classes were excluded).

**Questions:**

Why were Dirichlet-based approaches and other recent uncertainty quantification methods excluded from the comparison?
(See also comment on baseline comparisons).

Please elaborate on the loss function (10). Why is this a sensible choice? It is not immediately clear to me from the corresponding explanations in the paper.

How does the Interval SoftMax function handle cases where the intervals for different classes overlap significantly? What impact does this have on the final predictions and uncertainty estimates?

Instead of learning upper/lower probabilities, what is the difference to get the upper/lower probabilities e.g. from an ensemble of NNs?

How do you guarantee that the computation of the lower and upper probabilities within the credal set is scalable for high-dimensional classification tasks?

**Limitations:**

There is a very short discussion of potential limitations at the end of the paper.

---

> ### Author Rebuttal · Authors · 2024-08-07
>
> Dear Reviewer,
>
> We appreciate your efforts in reviewing our work and your recognition of our novelty. Below we address your concerns.
>
> **Response to W1:** We believe that we have provided a comprehensive evaluation of our approach, which consistently shows significantly improved performance, uncertainty quality, OOD detection, and model calibration in several popular and important setups. For instance, in Table 3 of our paper, we can observe the OOD detection improvements of about 5% and 11% for CIFAR 10 vs SVHN (AUROC) on VGG16 and ViT base architectures, respectively. Such visible improvements can be consistently observed in, for example, Table 2, Figures 5, 6, and Tables 8-12 of the paper.
>
> **Response to W2 and Q2:** We value the opportunity to provide a more rigorous argument of the theoretic analysis and loss design.
>
> Epistemic uncertainty in predictions is induced by the probability distribution(s) that generated the training data may be very different from the distribution that generates a test point. It is well known that using CE as a loss leads to the maximum likelihood estimator of the model parameters, under the assumption that all training points are drawn i.i.d. from a single, unknown distribution $P$:
> $$
> \hat{\theta} = \arg \min_\theta\text{CE} = \theta_{MLE}.
> $$
>
> What does Distributionally Robust Optimization (DRO) do? It takes a cautious stance, and assumes training points are generated by one of many potential distributions $\mathcal{U}$, and minimizes the upper bound of the expected risk (Eq. (6) of the paper). In practice, this is approximated using Adversarially Reweighted Learning as in Eq. (9). Therefore, it learns
> 			$\hat{\theta} = \arg \min_{P \in \mathcal{U}} E_P (Loss)$.
> However, if we apply DRO to a traditional network, the network will learn a single parameter value (Eq. (9) of the paper); for a new test data point $x_t$ it will then produce a single predicted probability vector, the one produced by the model that, in the long run, minimizes the worst-case risk.
>
> This, though, does not express at all the (epistemic) uncertainty about *which* of the distributions in $\mathcal{U}$ has generated the particular test datapoint $x_t$; as a result, it does not express how uncertain the prediction may be for that particular data point.
>
> Allowing a network to output lower and upper probability vectors for each input, instead, allows this epistemic uncertainty to be explicitly modeled.
>
> What is the best-case scenario? That in which all data (including the test point $x_t$) are indeed generated by the same distribution – in this case, $\theta_{MLE}$ is the best choice, in a maximum likelihood sense.
>
> What is the worst case? That in which $x_t$ is generated by a distribution that is far away from the groups of distributions $P_g$ (see lines 140-141) that generated the training data. In this case, the DRO solution will still be far from perfect, but still the best one can do, given the evidence in the training data.
>
> Therefore, if I minimize the CE versus $\boldsymbol{q}_U$ (first component of Eq. (10)), I get a credal set Cre of predictions whose upper probability is such that, for each class $i = 1,…,C$, there exists a probability vector $\boldsymbol{q}\in \mathbb{Q}$ such that $q(i)$ is the MLE prediction for that class (please remember the discussion of lines 172-175).
>
> Similarly, if I minimize the DRO loss versus $\boldsymbol{q}_L$ (the second component of Eq. (10)), for each class $i = 1,…,C$, there exists a probability vector $\boldsymbol{q} \in \mathbb{Q}$ such that $q(i)$ is the DRO prediction for that class.
>
> Minimizing the two components together, as in Eq. (10), thus encourages the predicted credal set to extend from the best-case to the worst-case prediction.
>
> **Response to W3 and Q1:** We kindly invite you to follow the detailed response in our global rebuttal (Part 2) for the comparison baselines.
>
> **Response to Q3:** Let us consider a numerical example in a three-element case.
> If we have three intervals such as $[1.1, 3.1]$, $[1, 3]$, and $[1, 2]$ as the inputs of the Interval SoftMax, the Interval SoftMax returns the following probability intervals as the outcomes: $[0.15612615, 0.71893494]$, $[0.12732467, 0.70053512]$, $[0.09810377, 0.43282862]$. We can observe that the Interval SoftMax also generates overlapped probability intervals to correctly reflect the overlap in the input. In extreme cases, If all the probability intervals over all classes are significantly overleaped, the resulting credal set could include the uniform probability. However, this does not prevent us from using the maximin and maximax criteria in Eq. (11) to derive a class prediction according to the conservativeness of the decision maker. Uncertainty estimation methods for the credal sets are also valid.
>
> **Response to Q4:** The main difference is that getting upper/lower probabilities e.g. from an ensemble of NNs would be a post-processing procedure only in the prediction space, no training process is involved. In our CreDEs, the model learns the probability interval from data and could encourage the predicted credal set to extend from the best-case to the worst-case prediction.
>
> **Response to Q5:** The Interval SoftMax Eq.(4) can ensure a valid credal set regardless of the dimension of the tasks. To reduce the computational complexity for high-dimensional classes, we performed Alg. 2 to reduce the dimension from high $C$ to $K$. Alg. 2 has a strong mathematical background in probability intervals [17] (Eq. (12) of the paper) and can rigorously guarantee a valid credal set. In addition, the ablation study in Appendix B.2 and the OOD detection cases for ImageNet in Table 1 show the effectiveness of the reduction in uncertainty quantification of credal sets.
>
> **Response to L1:** Following your comments, we will duly extend the discussion on the computational complexity and comparison baselines in the revised version, as we discussed in the global rebuttal.

---

> > ### Comment · Reviewer_AC8P · 2024-08-10
> > **Response to authors**
> >
> > I thank the authors for their effort responding to my questions.
> >
> > > Epistemic uncertainty in predictions is induced by the probability distribution(s) that generated the training data may be very different from the distribution that generates a test point.
> >
> > Can you please elaborate on this comment? If the test data is generated from a distribution different from the training data, the model may be uncertain because it hasn't been trained on similar data. While this is true in the OoD case, it's not the only factor contributing to epistemic uncertainty. If I am not wrong, the motivation for this paper is not only the OoD case, correct?
> >
> > > Minimizing the two components together, as in Eq. (10), thus encourages the predicted credal set to extend from the best-case to the worst-case prediction.
> >
> > This does not fully address my question. For clarification, let me ask a follow-up question, which will help me understand why Eq. (10) is a sensible choice:
> >
> > Why does Eq. (10) give incentive to learn suitable interval predictions? Are we not minimizing the first term in the sum in (10) by setting all upper probabilities to 1?
> >
> > > Response to Q4: The main difference is that getting upper/lower probabilities e.g. from an ensemble of NNs would be a post-processing procedure only in the prediction space, no training process is involved. In our CreDEs, the model learns the probability interval from data and could encourage the predicted credal set to extend from the best-case to the worst-case prediction.
> >
> > I apologize if my question was ambiguously stated. I see your point, that your approach involves a learning process instead of “post-processing” of the ensemble predictions. The question would be, can’t we achieve similar good results with this naïve post-processing approach? At least it would be interesting to look at.
> >
> > I remain skeptical about the loss itself, but given the results in the global rebuttal and taking into account the responses to my questions, I increased my initial score.

---

> ### Author Response · Authors · 2024-08-12
> **Further response to the remaining questions**
>
> Dear reviewer,
>
> We deeply appreciate your recognition of our response and your engagement in the discussion with us. In the following, we address your remaining questions.
>
> **Interpretation of the epistemic uncertainty**
>
> In supervised learning, we assume an underlying data-generating process in the form of an unknown probability distribution $P$ on the input space $\mathbb{X}$ and the target space $\mathbb{Y}$. The lack of knowledge regarding the underlying process is the source of the epistemic uncertainty [A, B, C]. In an ideal scenario, if the model could learn the exact distribution $P$ from the training data, there would be no epistemic uncertainty about a prediction given a new input. Nevertheless, in practice, given that the training dataset represents only a subset of the full space and that the test data may differ from the training data, it is not possible to guarantee that the distribution $\hat{P}$ learned from the training data will be identical to the true distribution $P$. Therefore, the epistemic uncertainty is induced. In our paper, we proposed CreDEs that output lower and upper probability vectors for each input to express this epistemic uncertainty in predictions.
>
> OOD data are significantly different from the training data, but not limited to the OOD data. For example, active learning aims to efficiently train models using minimal data by selectively acquiring additional samples, which are then labeled by experts. In this context, a larger EU indicates that such samples are significantly different from those used in the previous active learning iteration. Using the effective epistemic uncertainty (EU) measure as the acquisition function to obtain samples with a higher EU allows for an efficient active learning procedure [D].
>
> The motivation of our paper is to improve the uncertainty estimation using Credal Deep Ensembles, not limited to the OOD case. We chose OOD detection benchmarks as the main evaluation because they are widely used to assess the quality of epistemic uncertainty estimation. We also performed an active learning case study in Appendix B.5 of the paper, which also shows that our CreDEs provide improved uncertainty quantification.
>
> If we consider only the in-domain samples, effective uncertainty estimation can also improve classification performance through a rejection operation [C, E]. For example, when processing a batch of input data, those with higher epistemic uncertainty are first rejected, and then the accuracy of the remaining test samples is calculated. If the estimation of the prediction uncertainty is valid, the accuracy shows a monotonic increase. To illustrate, we show the results of the CreDEs and DEs on the CIFAR100 dataset in Table 1.
>
> *The result shows that our CreDEs provide valid EU estimates and better accuracy than DEs.*
>
> Table 1. Test accuracy on the CIFAR100 dataset using EU estimates as rejection metrics under different rejection rates.
> |       Rejection Rate       |         0        |        0.1       |        0.2       |        0.3       |        0.4       |        0.5       |        0.6       |        0.7       |        0.8       |        0.9       |      1.0      |
> |:--------------------------:|:----------------:|:----------------:|:----------------:|:----------------:|:----------------:|:----------------:|:----------------:|:----------------:|:----------------:|:----------------:|:-------------:|
> |            DEs-5           |   75.80 ± 0.28   |   80.48 ± 0.29   |   84.80 ± 0.29   |   88.37 ± 0.16   |   92.07 ± 0.15   |   95.28 ± 0.16   |   97.44 ± 0.23   |   98.73 ± 0.15   |   99.49 ± 0.10   |   99.87 ± 0.06   | 100.00 ± 0.00 |
> | CreDEs-5 ($\hat{i}_{min}$) | **79.54 ± 0.21** | **84.26 ± 0.23** | **87.84 ± 0.18** | **90.90 ± 0.22** | **93.80 ± 0.13** | **96.63 ± 0.15** | **98.32 ± 0.13** | **99.10 ± 0.10** | **99.62 ± 0.15** | **99.90 ± 0.09** | 100.00 ± 0.00 |
> | CreDEs-5 ($\hat{i}_{max}$) | **79.65 ± 0.19** | **84.31 ± 0.22** | **87.86 ± 0.18** | **90.90 ± 0.22** | **93.80 ± 0.13** | **96.63 ± 0.15** | **98.32 ± 0.13** | **99.10 ± 0.10** | **99.62 ± 0.15** | **99.90 ± 0.09** | 100.00 ± 0.00 |
>
> [A]  A review of uncertainty quantification in deep learning: Techniques, applications and challenges. Information Fusion, 2021
>
> [B] Aleatoric and epistemic uncertainty in machine learning: An introduction to concepts and methods. Machine Learning, 2021
>
> [C] Quantification of credal uncertainty in machine learning: A critical analysis and empirical comparison, UAI, 2022.
>
> [D] Deep deterministic uncertainty: A new simple baseline. CVPR 2023.
>
> [E] Benchmarking common uncertainty estimation methods with histopathological images under domain shift and label noise." Medical image analysis, 2023.

---

> ### Author Response · Authors · 2024-08-12
> **Further response to the remaining questions**
>
> **Ablation study on comparing our CreDEs and the post-processing approach**
>
> Thank you for your enlightenment. Inspired by your comments, we did an ablation study to compare the methods.
> Regarding the post-processing approach, we compute the upper and the lower bound per class $k$ from the five single predictions from DEs-5, as follows:
> $$
> q_{U_k} =  \max_{n=1,..,5}{q_{n,k}};  p_{L_k} = \min_{n=1,..,5}{q_{n,k}}.
> $$
>
> Then, we formulated a credal set by post-processing for uncertainty representation. Table 2 reports the performance comparison on CIFAR10 vs SVHN and Tiny-ImageNet benchmarks.
>
> Table 2: OOD performance comparison on CIFAR10 vs SVHN and Tiny-ImageNet benchmarks.
>
> |                           | SVHN: AUROC | SVHN: AUPRC | Tiny-ImageNet: AUROC | Tiny-ImageNet: AUPRC |
> |:-------------------------:|:-----------:|:-----------:|:--------------------:|:--------------------:|
> |           DEs-5           |    89.58    |    92.29    |         86.87        |         83.02        |
> |  DEs-5 (post processing)  |    93.77    |    96.06    |         88.78        |         86.83        |
> |  CreDEs-5 ($\delta=0.5$)  |  **96.55**  |  **98.17**  |         88.10        |         **87.85**        |
> | CreDEs-5 ($\delta=0.875$) |  **97.95**  |  **96.92**  |       **89.18**      |       **89.72**      |
>
> *From Table 2, we can observe that the post-processing approach can improve the performance of classical DEs.  Our method still performs better than others. In addition, due to the post-processing nature (representing the DEs predictions in credal sets), our CreDEs also achieve better performance in test accuracy and ECE values. For instance, test accuracy: **93.75%** ($\delta=0.5$), **93.87%** ($\delta=0.875$) vs 93.32% (DEs-5) and ECE: **0.0092** ($\delta=0.5$), **0.0093** ($\delta=0.875$) vs 0.0131.*
>
> We deeply thank your comments and inspiration, we believe incorporating more extensive comparisons, especially in real-world applications such as medical image analysis could strengthen our future work.
>
> **Following questions about the loss Eq. (10)**
>
> For a training batch, our CreNet first forward propagates via the Interval SoftMax activation (Eq. (4) of the paper) to generate meaningful upper and lower probability vectors to define the credal sets (lines 105-166). Because of the functional structure of Interval SoftMax activation, the upper and lower probability vectors are not computed independently but are correlated.
>
> In backward propagation, we apply the upper and lower probability vectors to the vanilla and DRO components, respectively. Following the rationale of the design choices (as we explained in lines 157-166 of the paper and the previous responses), the learned prediction interval can represent the boundary cases of the credal set by considering the optimistic scenario (Vanilla component) and the pessimistic case (DRO component), respectively. Namely, assuming that the true class of the label is index $j^*$, the Vanilla and DRO components in Eq. (10) minimize the Cross-Entropy loss $-log_2q_U(j*)$ and $-log_2q_L(j^*)$, respectively, in a mini-batch optimization process (lines 167-171). Thanks to Interval SoftMax, the trained probability intervals are valid for generating credal sets.
>
> In addition, because of the correlated computation between the upper and lower probability vectors in Interval SoftMax, minimizing the DRO component also affects the upper probability, driving the solution away from the all-one upper probability vectors.
>
> Once again, Thank you very much for your effort and your engaging in this discussion! We hope the increased clarity in our responses has addressed your concerns.

---

### Official Review · Reviewer_HFiw · 2024-07-13

**Soundness:** 3
**Presentation:** 3
**Contribution:** 3
**Rating:** 6
**Confidence:** 4

**Summary:**

In this paper, the authors propose a novel method for uncertainty estimation in deep networks called Credal Deep Ensembles, which combines credal inference and ensembling approaches. During inference, the model predicts intervals (lower and upper probability values) for each class, resulting in a final output that is a simplex subset defined by these intervals. The ensembling technique predicts N such intervals and averages them out. Based on this final prediction, the authors derive several uncertainty measures (both aleatoric and epistemic). The authors validate their method through extensive experiments on several image classification tasks (CIFARs and ImageNets) and various model architectures (ResNets, VGGs, ViTs), demonstrating significant improvements over ensembles in terms of model accuracy, quality of uncertainty estimations, OOD detection, and model calibration.

**Strengths:**

* The paper is clearly written and easy to follow. The idea is intuitive and easy to grasp. The related work section provides an adequate discussion of existing approaches to credal learning and nicely describes the proposed method in detail.

* The proposed methodology is interesting and provides a novel perspective on second-order approaches as evidential learning by combining credal inference with ensembling techniques.

* The authors provided an extensive evaluation of their approach, showing improved performance, uncertainty quality, OOD detection, and model's calibration in several popular and important setups.

**Weaknesses:**

* The approach involves too many parameters and procedures for training and inference, making it more complex compared to other methods such as ensembles. This added complexity could limit its practicality (and applicability) and ease of implementation.

* The paper does not provide a comparison with second-order methods such as evidential learning, which leaves a gap in the evaluation of the proposed approach. Including such comparisons could be important since the proposed credal set approach is deeply connected to second-order distribution prediction methods. Including such comparisons would strengthen the validation of the proposed approach and offer a more comprehensive assessment of its performance against established techniques.

* The output size is doubled, which, for tasks with a high number of classes, could significantly increase the number of weights. This could result in higher computational costs and memory requirements, potentially limiting the scalability and efficiency of the approach.

* It is not entirely clear where the improvements over ensembles are coming from or why credal inference brings additional benefits. More explanation is needed to understand the source of these enhancements.

* From the perspective of the experimental evaluation, I would be curious to see evidence that the behavior demonstrated in the paper would hold in other domains, such as texts, graphs, more complicated vision tasks (e.g. segmentation), not limiting to image classification task.

I tend to assess the paper positively and would be happy to increase my score after the discussion.

**Questions:**

* Given the increased complexity and number of parameters in the proposed approach compared to other methods like ensembles, how do the authors justify its practicality and ease of implementation? Are there any strategies to mitigate these complexities (for example, how to chose $\delta$ parameters for training or $K$ parameter for the optimized inference)?

* I believe that comparisons with and discussions of second-order methods such as evidential learning in the context of credal inference are important. How does the proposed approach perform relative to these techniques?

* The paper mentions significant improvements over ensembles but does not clearly explain why credal inference brings additional benefits. Can the authors provide a more detailed explanation of the source of these improvements?

**Limitations:**

The authors provide an adequate discussion of limitations in the last section.

---

> ### Author Rebuttal · Authors · 2024-08-06
>
> Dear Reviewer,
>
> We express our sincere gratitude for acknowledging our work and your valuable feedback. In the following, we address your concerns.
>
> **Regarding W4 and Q3: A more detailed explanation of the source of the improvements**
>
> **Response:** We would like to make a clearer explanation below.
>
> Epistemic uncertainty in predictions is induced by the probability distribution(s) that generated the training data may be very different from the distribution that generates a test point. It is well known that using CE as a loss leads to the maximum likelihood estimator of the model parameters, under the assumption that all training points are drawn i.i.d. from a single, unknown distribution $P$:
> $$
> \hat{\theta} = \arg \min_\theta\text{CE} = \theta_{MLE},
> $$
> where $\hat{\theta}$ denotes the learned parameters.
>
> What does Distributionally Robust Optimization (DRO) do? It takes a cautious stance, and assumes training points are generated by one of many potential distributions $\mathcal{U}$, and minimizes the upper bound of the expected risk (Eq. (6) of the paper). In practice, this is approximated using Adversarially Reweighted Learning as in Eq. (9). Therefore, it learns using the following equation:
> 			$$\hat{\theta} = \arg \min_{P \in \mathcal{U}} E_P (Loss)$$
>
> However, if we apply DRO to a traditional network, the network will learn a single parameter value (Eq. (9) of the paper); for a new test data point $x_t$ it will then produce a single predicted probability vector, the one produced by the model that, in the long run, minimizes the worst-case risk.
>
> This, though, does not express at all the (epistemic) uncertainty about *which* of the distributions in $\mathcal{U}$ has generated the particular test datapoint $x_t$; as a result, it does not express how uncertain the prediction may be for that particular data point.
>
> Allowing a network to output lower and upper probability vectors for each input, instead, allows this epistemic uncertainty to be explicitly modelled.
>
> What is the best-case scenario? That in which all data (including the test point $x_t$) are indeed generated by the same distribution – in this case, $\theta_{MLE}$ is the best choice, in a maximum likelihood sense.
>
> What is the worst case? That in which $x_t$ is generated by a distribution that is far away from the groups of distributions $P_g$ (see lines 140-141) that generated the training data. In this case, the DRO solution will still be far from perfect, but still the best one can do, given the evidence in the training data.
>
> Therefore, if I minimize the CE versus $\boldsymbol{q}_U$ (first component of Eq. (10)), I get a credal set Cre of predictions whose upper probability is such that, for each class $i = 1,…,C$, there exists a probability vector $\boldsymbol{q}\in \mathbb{Q}$ such that $q(i)$ is the MLE prediction for that class (please remember the discussion of lines 172-175).
>
> Similarly, if I minimize the DRO loss versus $\boldsymbol{q}_L$ (the second component of Eq. (10)), for each class $i = 1,…,C$, there exists a probability vector $\boldsymbol{q} \in \mathbb{Q}$ such that $q(i)$ is the DRO prediction for that class.
>
> Minimizing the two components together, as in Eq. (10), thus encourages the predicted credal set to extend from the best-case to the worst-case prediction.
>
> **Regarding Q1: any strategies to mitigate these complexities ($\delta$ parameters for training or $K$ parameter for the optimized inference)**
>
> **Response:** In our main evaluation, we set by default $\delta=0.5$ to reflect a balanced assessment of the train-test divergence and show how such a value allows our model to outperform the baselines. Most importantly, the ablation study on $\delta$ (Lines 287-312) demonstrates that our CreDEs are not particularly sensitive to $\delta$.
>
> One possible way to find the best $\delta$ in practice is to conduct standard cross-validation on specific test scenarios. Perspectively, an interesting option, in the presence of multiple datasets (e.g.acquired over time in a continual learning setting), could be applying the DRO loss component to different components of the training set, and assessing the results to robustly select $\delta$.
>
> $K$ is a trade-off parameter between uncertainty estimation and computational efficiency. The ablation study in Appendix B.2 shows that increasing the value of $K$ improves the OOD detection performance, but leads to an increase in execution time. Applying  $K<=10$ to the Generalized Hartly (GH) measure for EU estimation is more practical. Choosing $K$ for upper and lower entropy measures is more flexible. For instance, in Table 2 of the paper,  $K=20$ for the ImageNet dataset (original 1k classes) shows an improved OOD detection performance compared to DEs.
>
> **Regarding concerns about the complexity (W1 and W3) and the comparison baselines (W2 and Q2)**
>
> **Response:** We kindly invite you to follow the detailed response in our global rebuttal part 1 and part 2, respectively.
>
> **Regarding W5: Expecting evidence in other domains**
>
> **Response:** We fully agree with your comments that showing more evidence of performance improvement of our CreDEs in other domains can further strengthen our work. Given the consistent performance improvements of CreDEs on a wide range of image classification tasks (which you acknowledged as our strength), we are optimistic about successfully extending our work to other domains. We will include your suggestion in the discussion of our future work and find its proper place within the expanded scope of that work.

---

> > ### Comment · Reviewer_HFiw · 2024-08-12
> >
> > Thank you for thoroughly addressing my concerns in your rebuttal. I find the proposed method both novel and compelling, especially with the detailed explanations you provided regarding the source of the improvements and the strategies to manage the complexity of the approach. Additionally, your acknowledgment of the potential for extending this work to other domains strengthens the paper's broader applicability. Given these considerations, I find the contributions of this paper to be very interesting and impactful, and I would like to upgrade my score by 1 point.

---

> > > ### Author Response · Authors · 2024-08-13
> > >
> > > Dear Reviewer,
> > >
> > > Thank you for recognizing the value of our work and supporting the publication of our paper. We are glad that your concerns are fully addressed. Once again, we appreciate your time and effort in reviewing our work and providing valuable comments.

---

### Author Rebuttal · Authors · 2024-08-06

Dear Reviewers,

We appreciate your efforts in reviewing our paper and recognizing the novelty and strengths of our work. In the following, we would like to address your concerns regarding the complexity of our Credal Deep Ensembles (CreDEs) method and the comparison baselines.

**Part 1: added complexity compared to classical Deep Ensembles**

**Response:** We fully acknowledge the practical limitations of our method when there are strict computational resources (Lines 347-348). However, the extensive experiments show the scalability of our CreDEs on larger datasets (e.g., ImageNet) and larger network architectures (e.g., vision transformer). In addition, CreDEs consistently demonstrate visible improvement in a wide range of tasks and datasets, compared to deep ensembles (DEs). Therefore, CreDEs have great practical potential to enhance the performance of classical DEs in diverse real-world applications, such as medical image analysis [42], flood uncertainty estimation [10], structural health monitoring [70], or uncertainty-aware human motion forecasting [E].

Regarding inference time, doubling the final layer nodes would slightly increase the inference time. For instance, the inference time per sample for a ResNet50 architecture on the ImageNet dataset is 5.5 ms for a single standard neural network, vs 5.7 ms for a single CreNet (a marginal increase). The inference cost on the CIFAR10/100 dataset reported in *Table 1 of the rebuttal file* further demonstrates a slight complexity increase in our method.

Regarding the uncertainty estimation cost, we reported the cost of calculating the Generalized Hartley (GH) measure and the upper entropy in Figures 7 and 8 of the paper, respectively. For instance, the time cost for GH calculation for CIFAR10 without approximation is 17 ms (0.02 ms in the reduced case considering 4 out of 10 classes) while calculating the Epistemic Uncertainty (EU) in deep ensembles for CIFAR10 takes $4.1\times 10^{-4}$ ms, measured on the same single CPU. Though higher CreDEs remain practical without actual computational constraints. Besides, the numbers reported are without code efficiency optimization: a more efficient code implementation could significantly reduce the cost.

Regarding the training complexity, the CreNet uses a custom training loop, unlike the TensorFlow-standardized training standard neural networks, precluding a fair comparison. Given the evidence that CreNets only marginally increases the inference time (single forward pass) in Table 6, we are optimistic that by standardizing and optimizing the custom training loop and adopting a more efficient code implementation of Algorithm 1, the training effort could be significantly reduced.

**Part 2: Comparison limited to DEs**

**Response:** We chose DEs for comparison, as they serve as a strong baseline for uncertainty estimation, e.g., shown in studies [A, B, C].  More recently, study [A] shows that DEs are good choices across the board for uncertainty estimation in their evaluation. Through extensive evaluation, we draw our main conclusion: CreDEs show significantly and consistently improved performance, uncertainty quality, OOD detection, and model calibration in several popular and important setups. We much appreciate the reviewers' acknowledgment of the extensive evaluation as one of the strengths of our paper.

The main reason for excluding Bayesian neural network (BNN) approaches in our original paper is that they generally have difficulty scaling to large datasets and complex network architectures [C]. Inspired by the suggestions, we conducted an additional comparison between CreDEs and DEs, MCDropout [F], and two TensorFlow-standardized BNNs (BNN-R [G]  and BNN-F [H]). All the models are trained on the ResNet50 for the CIFAR10 dataset from scratch. The Adam optimizer is applied with a learning rate scheduler, initialized at 0.001. The learning rate is subject to a reduction of 0.1 at epochs 80 and 120. For BNNs, 10 forward passes are used for uncertainty estimation. The uncertainty evaluation via OOD detection on the CIFAR10 vs SVHN/Tiny-ImageNet dataset is reported in *Table 2 of the rebuttal file*. The results consistently demonstrate the significant improvements of our methods.

We excluded the Dirichlet-based approaches (lines 69-76) (other branches of second-order models) mainly because, they are generally trained to predict Dirichlet distributions from only one-hot label data. A criticism of these methods is the lack of Dirichlet distribution labels in the training process, and the performance of the models often deviates from theoretical EU assumptions [69]. In addition, a recent study [D] showed that epistemic uncertainty is generally not faithfully represented in these methods and the resulting measures of epistemic uncertainty cannot be interpreted quantitatively.

We will duly extend the above discussion in the Limitation and Future work in the revised version. One of the essential objectives of our future work is to build a universal uncertainty evaluation framework, and comprehensively assess our methods together with various other models. We believe that the kind of broader comparison you suggest will find its right place within the expanded scope of that work.

[A] Benchmarking uncertainty disentanglement: Specialized uncertainties for specialized tasks. 2024.

[B] Deep ensembles work, but are they necessary? NeurIPS 2022

[C] Deep deterministic uncertainty: A new simple baseline. CVPR 2023

[D] Is Epistemic Uncertainty Faithfully Represented by Evidential Deep Learning Methods? ICML 2024

[E] De-tgn: Uncertainty-aware human motion forecasting using deep ensembles. IEEE Robotics and Automation Letters 2024

[F] Dropout as a Bayesian Approximation: Representing Model Uncertainty in Deep Learning, ICML 2016

[G] Variational dropout sparsifies deep neural network. ICML 2017

[H] Flipout: Efficient pseudo-independent weight perturbations on mini-batches. iCLR 2018

---

### Decision · Program_Chairs · 2024-09-25

**Decision:**

Accept (poster)

**Comment:**

Dear reviewers, thank you for all the hard work you put in your reviews and in engaging with the authors and their responses.
To the Authors, thank you for giving such a detailed rebuttal.

After reading all of the reviews and comments, I am very happy to recommend this paper for acceptance. Especially after the authors in depth rebuttal.

Best,

    AC